# Targeted Therapies for Hepatocellular Carcinoma Treatment: A New Era Ahead—A Systematic Review

**DOI:** 10.3390/ijms232214117

**Published:** 2022-11-15

**Authors:** Christos Damaskos, Nikolaos Garmpis, Dimitrios Dimitroulis, Anna Garmpi, Iason Psilopatis, Panagiotis Sarantis, Evangelos Koustas, Prodromos Kanavidis, Dionysios Prevezanos, Gregory Kouraklis, Michail V. Karamouzis, Georgios Marinos, Konstantinos Kontzoglou, Efstathios A. Antoniou

**Affiliations:** 1Renal Transplantation Unit, Laiko General Hospital, 11527 Athens, Greece; 2Nikolaos Christeas Laboratory of Experimental Surgery and Surgical Research, Medical School, National and Kapodistrian University of Athens, 11527 Athens, Greece; 3Second Department of Propedeutic Surgery, Laiko General Hospital, Medical School, National and Kapodistrian University of Athens, 11527 Athens, Greece; 4First Department of Propedeutic Internal Medicine, Laiko General Hospital, Medical School, National and Kapodistrian University of Athens, 11527 Athens, Greece; 5Charité—Universitätsmedizin Berlin, Corporate Member of Freie Universität Berlin and Humboldt—Universität zu Berlin, Augustenburger Platz 1, 13353 Berlin, Germany; 6Molecular Oncology Unit, Department of Biological Chemistry, Medical School, National and Kapodistrian University of Athens, 11527 Athens, Greece; 7Medical School, National and Kapodistrian University of Athens, 11527 Athens, Greece; 8Department of Hygiene, Epidemiology and Medical Statistics, Medical School, National and Kapodistrian University of Athens, 11527 Athens, Greece

**Keywords:** targeted, therapy, hepatocellular, carcinoma, cancer, hepatocarcinogenesis

## Abstract

Hepatocellular carcinoma (HCC) remains one of the most common malignancies and the third cause of cancer-related death worldwide, with surgery being the best prognostic tool. Among the well-known causative factors of HCC are chronic liver virus infections, chronic virus hepatitis B (HBV) and chronic hepatitis virus C (HCV), aflatoxins, tobacco consumption, and non-alcoholic liver disease (NAFLD). There is a need for the development of efficient molecular markers and alternative therapeutic targets of great significance. In this review, we describe the general characteristics of HCC and present a variety of targeted therapies that resulted in progress in HCC therapy.

## 1. Introduction

Hepatocellular carcinoma (HCC) is the most common primary liver malignancy worldwide. Thus, it remains a significant health problem across the world, especially in sub-Saharan African and Asian countries [1]. HCC is the fifth most frequent cancer diagnosis among male patients and the sixth most frequent cancer diagnosis in female patients. HCC results in between 250,000 and 1,000,000 deaths worldwide. The incidence of liver cancer varies widely according to geographic location. The causative factors of HCC have been thoroughly examined and described: chronic hepatitis virus B (HBV) and chronic hepatitis virus C (HCV), liver cirrhosis, non-alcoholic fatty liver disease (NAFLD), tobacco smoking, and aflatoxin consumption. A sex preponderance is noted with a two to eight times higher incidence rate in male patients, given the significantly higher incidence of chronic virus liver infections in male patients worldwide. Moreover, tobacco smoking and heavy alcohol consumption are common in male patients, whereas a possible protective role of estrogens is also being investigated [2,3,4].

Orthotopic liver transplantation (OLT) is the preferred treatment option for patients in a terminal stage. This kind of treatment results in one-year survival rates of 80–90% [5]. The development of alternative treatment methods for patients in a terminal stage is crucial. Multiple surveillance programs have been designed for patients with a high risk of HCC in the developed world [6]. Nevertheless, due to limited resources in the underdeveloped countries, HCC tends to present in later stages, where curative treatment is not a feasible option [7].

A wide variety of imaging modalities are available to physicians to evaluate the nature of a liver tumor. The best modality for liver imaging remains a subject of debate. Ultrasonography (US) is a relatively inexpensive method of screening for HCC. It excels over both computer imaging (CT) scans and magnetic resonance imaging (MRI) scans with its lower cost and with no exposure to radiation or the possibility of the nephrotoxic actions of the contrast agents used in CT scans. US as a screening method for patients with chronic liver disease (liver cirrhosis) and chronic liver virus infections has been proven to have 60% sensitivity and over 95% specificity [8].

When a liver nodule has features suggestive of HCC, further diagnostic investigation is rendered necessary. Biopsy is indicated for patients with confirmed HCC with a liver nodule greater than 2 cm and low serum alpha-fetoprotein (AFP), or in patients in whom resection, ablation, or liver transplantation is not a feasible option [9]. Of note, in patients with elevated AFP and imaging studies indicative of HCC, biopsy should be avoided, and patients should be treated as having HCC according to the guidelines [10].

The molecular pathways and the histopathological changes that lead to HCC initiation are not fully understood yet. There are indications suggesting that genetic changes in pre-neoplastic hepatic cells and the gradual accumulation of mutations of the same cells are the cause of the malignant transformation that leads to HCC development [11,12].

It is well established that HCC can occur within a single liver nodule or in multiple liver nodules simultaneously. Furthermore, HCC is staged by pathologists as: well, moderate, and poorly differentiated. Well-differentiated HCC consists of cancer hepatic cells with similar characteristics to healthy hepatic cells, while in poorly differentiated HCC, cancer cells have different structures and characteristics compared to normal hepatic cells [12,13,14,15,16,17,18,19,20,21,22,23,24,25,26,27,28,29,30,31,32,33,34,35,36,37,38,39,40,41,42,43,44,45,46,47,48,49,50,51,52,53,54,55,56,57,58,59,60,61,62,63,64].

### 1.1. Liver Lobule: The Organization of Hepatic Parenchyma

The organization of hepatic parenchyma into microscopic functional units has been referred to as liver lobule. A liver lobule consists of a terminal hepatic venule at its center, which is surrounded by up to six terminal portal triads, all of which compose a polygonal unit. This polygonal unit is lined at the periphery of the liver lobule between each terminal portal triad by terminal portal triad branches. Between the terminal hepatic venule, at the center of the liver lobule, and the terminal portal trial branches, hepatocytes are located in a one-cell-thick layer and surrounded on each of their sides by endothelial-lined and blood-filled sinusoids. In this arrangement, blood flows to the terminal hepatic venule from the blood-filled sinusoids. Hepatic cells of each liver lobule produce bile, which is delivered to terminal canaliculi, formed on the lateral walls of the intercellular hepatic cells. Terminal canaliculi form bile ducts, which exude into the portal triads [17,18,19,20].

A wide variety of cells are found along the sinusoidal lining, including hepatic stellate cells, lymphoid cells, and Kupffer cells. Kupffer cells are specialized macrophages located in the liver. They act as phagocytes and have the ability to migrate across the liver sinusoids. In addition, Kupffer cells act as initiative factors for the inflammatory response, and the recognition of foreign substances along the sinusoids is one of their roles. Kupffer cells are not the only cell type of the hepatic cell defensive system. Other lymphoid cells such as CD^4+^ and CD^8+^ T-cells, natural killer T-cells (NKT), and natural killer cells are present as well. Liver stellate cells consist of a great amount of retinoid found in the space of Disse. These stellate hepatic cells have dendritic processes between hepatic cell microvilli, and they are wrapped around the endothelium of the liver sinusoids. Stellate cells have a major role in acute and chronic liver injuries initiating the development of liver fibrosis to cirrhosis [21,22].

The exposure to toxic substances and induction of immune responses in the liver can result in inflammation through Kupffer and stellate cells. This can lead to necrosis [23]. Inflammation of the liver parenchyma can lead to liver fibrosis and cirrhosis eventually. Liver cirrhosis is the most advanced stage of liver fibrosis. Moreover, distortion of the anatomy of the liver cells, septa, and nodule formation, as well as alterations to the normal blood flow within the liver lobules, are observed. Cirrhosis is well established as one of the main causative risk factors for HCC development [23,24]. While the cancer-triggering molecular mechanism of the cirrhosis is yet to be found, the constant process of liver cell necrosis and regeneration with increased cell turnover leads the hepatic cells to have a higher exposure to mutagenic agents. During this process, genetic and epigenetic changes are likely to happen. As a result, normal hepatic cells degenerate to dysplastic foci, liver nodules, and eventually HCC [25,26].

### 1.2. Staging Systems

On CT scans, HCC generally appears as a focal nodule with early enhancement on the arterial phase and rapid washout of the contrast on the portal vein phase on a three-phase contrast study. On the other hand, HCC appears as a high-signal-intensity lesion on MRI scans [27].

The prognosis of HCC is associated with both the underlying liver disease and the tumor characteristics [28]. As a result, multiple systems have been proposed for staging HCC.

The TNM is a staging system that has been proposed by the American Joint Committee of Cancer (AJCC) for solid tumors; the current edition is the 7th for HCC. TNM stands for: tumor, lymph nodes, and metastasis. While TNM staging reflects the prognosis of the disease in patients undergoing tumor resection, it cannot be used in the management of the disease, as it fails to measure the liver function. However, the tumor characteristics are predictive of the final outcome [10,28].

To date, the management of HCC has been based on the standard staging system proposed by the Barcelona Clinic Liver Cancer (BCLC). The BCLC staging system correlates the possible treatment options with patient outcome. In this system, stage 0 patients are characterized by a tumor smaller than 2 cm, normal portal pressure, and normal serum bilirubin. In these patients, resection of the tumor is a feasible and safe option with long-term survival rates that exceed 75%. Patients with a larger tumor, a single tumor smaller than 5 cm, or more than one tumor, none of which is over 3 cm, are eligible either for liver transplantation or surgical excision, since they have decompensated cirrhosis or preserved liver function, respectively [29,30,31].

Current treatment trends in HCC include liver resection and orthotopic liver transplantation as the gold standard procedures. Multiple techniques of hepatectomies have been described, and hepatic surgery is assisted by ablation techniques combined with transcatheter devices. Liver transplantation is effective for the patients with HCC who have a tumor not larger than 5 cm or up to three lesions with each 3 cm or smaller, with a 5-year overall survival (OS) rate of 75% [5]. Systemic and palliative therapies are the most common in the management of HCC due to the disease’s presentation in later stages most of the time [32,33]. HCC has proven to be resistant to chemotherapy [34]. As a result, targeted therapies have increased in popularity as a new weapon against HCC [35,36].

### 1.3. Expression of Growth Factors

HCC vascularity is characterized by structural and functional abnormalities that cause a tumor. Vascular incretion should be present in order for HCC cells to proliferate. Pro-angiogenic growth factors, as well as their receptor tyrosine kinases (RTK), participate significantly in angiogenesis since their activation in the microenvironment of the tumor mediates the initiation of tumor neoangiogenesis [37]. Various factors such as vascular endothelial growth factor (VEGF), epidermal growth factor (EGF), fibroblast growth factor (FGF), insulin-like growth factor (IGF), and platelet-derived growth factor (PDGF) seem to be responsible for high vascularity and cancer cell proliferation. These growth factors are not only expressed in cancer cells but also in healthy cells surrounding the tumors. It has been reported that the expression of these factors correlates with the expansion of the disease, as well as the vascular invasion [12,38,39,40].

RTK are transmembrane proteins that are responsible for transferring extracellular signals to the intracellular microenvironment, carrying molecules (such as glucose) from one side of the plasma membrane to the other. RTK are activated upon growth factor binding to their specific extracellular domain through dimerization and autophosphorylation of their receptor [41,42]. The interaction of the phosphorylated receptor with various cytoplasmic signaling molecules, such as the PI3K/AKT/mTOR and RAS/RAF/MEK/ERK pathways, results not only in angiogenesis but also in other processes such as cell survival and the migration, proliferation, and differentiation of endothelial cells.

Considering the fact that RTK are regulated at multiple levels, their dysregulation results in the transformation of the cell. The mechanisms involved—such as genomic rearrangements, gain of function mutations or deletions, over-expression of RTK, permanent stimulation of RTK from over-expressed growth factors, and retroviral transduction of a protooncogene—have been reported. All of them result in constantly active kinase activity [42,43,44,45].

RAS and RAF participate in intracellular signals which activate the expression of various genes [46,47]. At first, RAS activates RAF, which causes the activation of MEK [48,49]. Moreover, MEK activates not only ERK but also its phosphorylation [50,51]. ERK regulates various intracellular substrates directly and gene expression indirectly, such as cell kinase, in order to activate transcription factors and cell cycle regulators [51,52]. Finally, the activation of ERK is related to cancer cell proliferation.

Moreover, the expression of EGF in tumors has been reported to be related to tumor invasion [53]. Conversely, the expression of PDGF in tumors has been reported to be responsible for metastasis. Endothelial cell proliferation participates significantly in tumor infiltration to healthy parenchyma, as well as vascular invasion [54].

Concerning the phosphatidylinositol-3 kinase (PI3K) pathway, it plays an important role in the proliferation and survival of cancer cells in a variety of solid tumors, such as HCC [55]. It has been reported that PI3K activates the lipid second messenger AKT. Afterward, AKT phosphorylates various intracellular proteins, including mTOR [46]. The activation of mTOR propagates cell proliferation and inactivates BCL2-associated agonist of cell death (BAD). The inactivation of BAD is related to cancer cell survival by regulating apoptosis [56]. Moreover, the inactivation of AKT has been reported to have an antitumor effect, and thus it could be used in potential HCC treatment [57]. The regulation of the PI3K pathway by phosphatase and tensin homolog on chromosome 10 (PTEN) has been reported, as well as the suppression of the expression of PTEN in half of the HCC cells clinically [58]. In fact, PTEN expression is suppressed by HBx protein in HBV patients [59]. The importance of the downregulation of PTEN can be understood since it is associated with tumor grade progression, tumor stage, and poor overall prognosis [58,60,61,62].

Tyrosine kinase-type receptors are correlated with HCC development. It has been reported that VEGF receptors (VEGFR), PDGF receptors (PDGFR), EGF receptors (EGFR), FGF receptors (FGFR), and IGF receptors (IGFR) are responsible for initiating the intracellular RAS in the RAF/MEK/ERK pathway [59,63,64,65,66]. It is also known that AP-1 proteins, such as c-JUN and c-FOS, activate the expression of a wide range of genes responsible for cell proliferation and high vascularity [67]. The activation of the RAF/MEK/ERK pathway causes progression of HCC [68] and HBV-related HCC development [69]. Moreover, HCV proteins activate RAF and are considered to participate in the development of HCC [49,70,71,72,73,74].

Among these multiple signaling pathways, the one that is mediated by VEGF and VEGFR is one of the most investigated. VEGF actually refers to the VEGF-A isoform. Four spliced isoforms of VEGF-A have been reported (VEGF121, VEGF165, VEGF189, and VEGF206), with VEGF165 being the most predominant form. VEGF isoforms are released by normal and tumor cells in response to various stimuli, and they bind to VEGFR-1 and VEGFR-2, which are located on the cell surface of vascular endothelial cells and bone marrow-derived cells. VEGFR-3 is also a member of the same family as RTK; it is involved in lymphangiogenesis, but it is not a receptor for VEGFA but for VEGFC and VEGFD. The role of VEGFR-1 in the vascular endothelium is not completely understood yet, whereas VEGFR-2 is considered to be the major mediator of EC mitogenesis, survival, and angiogenesis, and microvascular permeability [75]. VEGF-A165 is commonly over-expressed by a variety of human tumors, such as colorectal cancer and gastric carcinoma, and its over-expression is associated with not only progression, invasion, and metastasis but also poorer survival and prognosis in patients [75,76,77,78]. Hence, a range of anti-angiogenetic agents have been developed, with a focus on targeting VEGF-A and VEGFR-2 [79].

However, none of the involved pathways was proved to be dominant. That is why a variety of drugs that target each pathway separately have been evaluated for HCC. Moreover, it is well established that HCC is a highly resistant malignancy to systematic chemotherapy. As a result, the current focus in treating HCC is to identify the signal pathways and the genes related to carcinogenesis as well as to chemotherapy resistance [80,81].

## 2. Materials and Methods

A search was conducted in MEDLINE (via PubMed) in order to retrieve relevant articles. The search terms employed were: hepatocellular carcinoma, targeted therapy, growth factors, tyrosine kinase inhibitors, immunotherapy, and clinical trials. Furthermore, we checked the references of all articles found aiming to include any other eligible studies. The PRISMA approach was used for the selection of the articles included in the review. A total of 1077 records in English were identified. Following removal of the duplicates, 951 records remained. The remaining articles were screened and 730 were excluded for various reasons. Some of them were only abstracts or reviews, whereas others were not completely relevant to the topic. It should be mentioned that publications in non-English languages were also excluded. The number of full-text articles assessed for eligibility was 221, and 206 full-text articles were included. The inclusion process is demonstrated in Figure 1.

## 3. Results

### 3.1. Multi-Targeted Tyrosine Kinase Inhibitors: First-Line Treatment

#### 3.1.1. Sorafenib

Sorafenib constitutes the first targeted therapy approved by the American Food and Drug Administration (FDA) for advanced HCC [82,83,84,85,86,87]. It is a small oral molecular agent that simultaneously affects more than one target. Specifically, it affects both the RAF/MEK/ERK pathway by inhibiting Raf-1 and the RTK that mediate the angiogenesis and progression of the tumor (Figure 2A) [88]. The Sorafenib Hepatocellular Carcinoma Assessment Randomized Protocol (SHARP) trial led to the approval of sorafenib by the US FDA. In detail, it was a phase III, double-blind, placebo-controlled study which randomly assigned 602 patients with advanced-stage HCC who had not previously received systemic treatment of either sorafenib (400 mg twice daily) or placebo. The eligibility criteria were the following: an Eastern Cooperative Oncology Group Performance Status (ECOG-PS) of 0–2, Child–Pugh liver function class A, adequate hepatic and hematologic function, and a life expectancy of 12 weeks or more. The results showed an improvement in median OS from 7.9 to 10.7 months (hazard ratio (HR) = 0.69, *p* = 0.001). Similarly, the time to progression (TTP) radiologically was also extended from 2.8 to 5.5 months (HR = 0.58, *p* < 0.001). The overall response rate (ORR) was 2%. The disease control rate (DCR) was significantly higher in the sorafenib group than in the placebo group (43% vs. 32%, *p* = 0.002) [83].

Another phase III trial conducted in Asia with 271 patients with similar eligibility criteria confirmed the findings of the SHARP trial. An increased OS in the group treated with sorafenib was noticed (6.5 vs. 4.2 months, HR = 0.68, *p* = 0.014). Patients in the same group also had a longer TTP (2.8 vs. 1.4 months, HR = 0.57, *p* = 0.0005). The partial response was 3.3% (vs. 1% in the placebo group), and the DCR was also greater (53% vs. 12%, *p* = 0.0019) [84].

Subsequent studies used data from the SHARP trial and further analyzed them in order to determine whether baseline patients’ characteristics can affect the efficacy and the safety of sorafenib. Bruix et al. selected five subgroup domains for analysis: etiology of HCC, tumor burden, extrahepatic spread, ECOG performance status, and tumor stage based on the BCLC system and treatment received before the use of sorafenib. The results showed that sorafenib improves OS and DCR independently of these characteristics [89]. Based on these data, sorafenib was established as a standard therapy for patients with advanced HCC.

Sorafenib has been also studied as part of first-line combination therapies [90,91,92]. Sorafenib 400 mg twice a day for 2 weeks, followed by concurrent mFOLFOX (5-fluorouracil (5-FU) 1200 mg/m^2^/day for 46 h, leucovorin 200 mg/m^2^, and oxaliplatin 85 mg/m^2^ biweekly) was studied in a phase II trial including 40 patients that showed that median TTP was 7.7 months (95% confidence interval (CI): 4.4–8.9 months), the median OS was 15.1 months (95% CI: 7.9–16.9), and the total DCR was 69%. However, moderate toxicity was caused. About three out of four patients required a dose reduction either of sorafenib or of mFOLFOX, and AST, ALT, and bilirubin elevation were among the grade ≥3 adverse events (AEs). Consequently, this combination showed promising efficacy against advanced HCC, but it would be wise to use it only in certain patients without hepatic dysfunction [91].

Sorafenib has also been studied in combination with transarterial chemoembolization (TACE) [93]. The START study was a phase II, open-label, prospective, single-arm, multicenter trial that evaluated the safety and efficacy of sorafenib plus TACE combination. TACE was performed by selective transarterial chemotherapy in the vessels feeding the tumor using an emulsion of lipiodol (5–20 mL) and doxorubicin (30–60 mg) followed by embolization with absorbable particles (gel foam). Patients received interrupted dosing of 400 mg BID sorafenib. Sorafenib was interrupted 4 days before TACE and was recontinued 4 days afterwards. The response was evaluated with a CT scan and AFP measurement. TACE cycles were repeated every 6–8 weeks when needed. A total of 192 patients with HCC and Child–Pugh A (91.8%) or B (7.1%) were analyzed. The safety profile was quite favorable, possibly due to the interrupted dosing of sorafenib. The DCR was 93.7%, the median progression-free survival (PFS) 384 days (95% CI: 322–469), the TTP 415 days (95% CI: 338–491), and the estimated 3-year OS rate 86.1% [92]. Another study that compared TACE vs. TACE plus sorafenib showed a prolongation of survival in patients receiving combination therapy. Among 118 patients with HCC and Child–Pugh A or B, 59 were treated with TACE and 59 with TACE followed by sorafenib, which was initiated one week after TACE. The DCR was 86.4% (51/59) in the sorafenib + TACE group and 67.8% (40/59) in the TACE group. The mean OS was 25.3 ± 2.6 months in the sorafenib + TACE group and 22.5 ± 2.5 months in the TACE group. The 1-, 2-, and 3-year survival rates were 71.9% (41/57), 42.1% (24/57), and 17.5% (10/57), respectively, in the sorafenib + TACE group and 52.6% (30/57), 31.6% (18/57), and 8.8% (5/57), respectively, in the TACE group. Both the DCR and OS in the sorafenib + TACE group were significantly superior to those in the TACE group (*p* = 0.027, *p* = 0.030) [94].

On the other hand, in a phase III trial including patients who had responded to TACE and also received sorafenib, no prolongation of TTP was found. After TACE, 458 patients were randomized to sorafenib or placebo. The period between TACE and the initiation of sorafenib was not short, with >50% starting it >9 weeks after TACE. The median TTP in the sorafenib and placebo groups was 5.4 and 3.7 months, respectively (HR = 0.87; 95% CI: 0.70–1.09; *p* = 0.252). No significant difference was observed in OS [95].

Thus, sorafenib also has encouraging efficacy as part of combination therapies, but more studies are needed to establish such practice and determine the exact combination.

#### 3.1.2. Lenvatinib

Lenvatinib is an oral multi-kinase inhibitor (MKI) that suppresses tumor cell proliferation and tumor angiogenesis via the inhibition of VEGFR1-3, FGFR1-4, PDGFR alpha, RET protein, and c-Kit protein (Figure 2B) [96]. Considering the activity that lenvatinib was showed to have in HCC in a phase II trial [97], a phase III non-inferiority trial (REFLECT) was designed, aiming to compare OS with lenvatinib vs. sorafenib as a first-line treatment in 954 assigned patients with advanced HCC. All the patients evaluated had BCLC stage B or C, Child–Pugh liver function class A, and a score of 0 or 1 in the Eastern Cooperative Oncology Group (ECOG) performance status. Group A was treated with lenvatinib (12 mg/day for bodyweight ≥60 kg or 8 mg/day for bodyweight <60 kg), and group B was treated with sorafenib (400 mg twice daily). All baseline characteristics were similar between the two groups, apart from baseline HCV etiology (19% in group A vs. 26% in group B) and AFP concentrations. The results showed a median OS of 13.6 months for lenvatinib versus 12.3 months for sorafenib (HR = 0.92, 95% CI: 0.79–1.06), a TTP of 8.9 months versus 3.7 months (HR = 0.63, 95% CI: 0.53–0.73, *p* < 0.0001), and superiority in response rate (24.1 vs. 9.2%, *p* < 0.001) compared with sorafenib [98]. Additionally, Briggs et al. reanalyzed the REFLECT study, suggesting that the favorable effect of lenvatinib had possibly been underestimated mainly because of imbalances between the two groups regarding the AFP concentrations and the use of additional treatments in the group of sorafenib [99].

#### 3.1.3. Sunitinib

Sunitinib is a multi-targeted tyrosine kinase inhibitor (TKI) with anti-angiogenic and antitumor activities. It targets VEGFR-1-3, PDGFR (alpha and beta), c-KIT, FLT3, and RET (Figure 2C). Its activity in HCC was first assessed in three phase II studies in which patients with advanced HCC were evaluated. The first was conducted by Faivre et al. More specifically, 37 patients were enrolled and treated with 50 mg/day for 4 weeks, followed by 2 weeks off treatment, and the primary endpoint was ORR. This study did not meet its primary endpoint, as sunitinib in this dose showed pronounced toxicities [100]. In the second one, 34 patients were enrolled and treated with 37.5 mg/day for 4 weeks, followed by 2 weeks off treatment, and PFS was set as the primary endpoint. According to the analysis, median PFS was 3.9 months (95% Cl: 2.6–6.9 months) and the 3-month PFS rate was 56%, whereas the targeted rate was 59%. In addition, sunitinib seemed to be well tolerated by most patients when it was used according to this dose schedule. Overall, this study managed to provide evidence of the antitumor activity of sunitinib [101]. In the third one, 44 patients were evaluated after receiving 37.5 mg sunitinib daily until progression or unacceptable toxicity. The primary endpoint was PFS at 12 weeks (PFS12). If ≤13 patients remained progression-free and alive after 12 weeks, the treatment would be considered uninteresting, otherwise it would be considered promising for further investigation. According to the results, PFS12 was rated as a success in 15 patients (33%, 95% CI: 20%–47%), and thus the therapy was considered promising [102]. Based on these outcomes, a randomized phase III trial was designed in order to compare sunitinib to sorafenib as a first-line treatment. More specifically, 1074 patients with advanced HCC were evaluated in total. This trial, however, did not achieve its primary endpoint as the OS with sunitinib was significantly inferior to that with sorafenib (7.9 vs. 10.2 months, HR = 1.30, *p* = 0.0014). Furthermore, sunitinib was associated with more frequent and severe side effects (including grade 3/4 thrombocytopenia (30%), neutropenia (25%), and hemorrhagic events (12%)), a fact that led to the premature closure of the study due to safety reasons. Hence, this trial failed to prove the superiority of sunitinib as a first-line treatment in patients with advanced HCC [103].

Additionally, other studies tried to evaluate the efficacy of combination therapies such as sunitinib plus TACE [103,104,105]. In a propensity score matching study including 103 patients with advanced HCC, 38 received sunitinib plus TACE and 65 only TACE. The primary endpoint was OS, while secondary endpoints included the TTP and AEs related to therapy. Sunitinib plus TACE was shown to have better results regarding OS and TTP when compared to TACE alone (OS: 8.8 vs. 6.3 months, respectively; *p* = 0.029/TTP: 3.9 vs. 2.5 months, respectively; *p* = 0.002). These results were also confirmed after selective matching by propensity scores and no important differences were noted regarding the AEs related to therapy [104].

Another prospective phase II trial, including 16 patients, studied the efficacy and safety of sunitinib plus TACE and demonstrated promising efficacy and acceptable toxicity. In fact, the median PFS was 8 months and OS 14.9 months [105].

Nonetheless, Hu et al. in a study including 104 patients with inoperable stage III HCC failed to prove the superiority as well as the non-inferiority of the combination of sunitinib plus TACE versus sorafenib plus TACE. A total of 51 patients received sunitinib plus TACE, while 53 received sorafenib plus TACE, and no statistically important differences were found between the characteristics of the two groups. However, both the OS and the PFS were significantly higher in the group treated with sorafenib plus TACE. Specifically, the median OS was 13.2 months (range 1.5–32.9) versus 9.2 months (range 1.3–28.2), and the median PFS 10.2 months (range 1.1–31.6) versus 6.9 months (range 0.7–29.6) [106].

#### 3.1.4. Linifanib

Linifanib is an oral TKI with activity against VEGFR and PDGFR (Figure 2D) [67]. A randomized phase III trial was designed in order to compare linifanib to sorafenib as a first-line treatment in patients with advanced HCC without previous systemic treatment. In total, 1035 patients were evaluated, of which 514 were assigned to linifanib 17.5 mg once daily and 521 to sorafenib 400 mg twice daily. The TTP and ORR favored linifanib. Unfortunately, these improvements did not manage to improve the outcome of the OS. OS (primary endpoint) was 9.1 months with linifanib and 9.8 months with sorafenib, and thus this trial failed to prove the superiority of linifanib compared to sorafenib. Moreover, the AEs with linifanib were experienced in a greater portion of patients than those receiving sorafenib, and these AEs led to discontinuations and dose reductions/interruptions, indicating that linifanib was less tolerated than sorafenib [59].

#### 3.1.5. Erlotinib

Erlotinib is an oral inhibitor of EGFR tyrosine kinase (Figure 2E) [49]. EGFR has been identified as a parameter promoting resistance of HCC cells to sorafenib. The inhibition of EGFR led to increased efficacy of sorafenib. This was proved by Ezzoukhry et al. who used model cell lines (Huh7, Hep3B, HepG2) in order to analyze the response of HCC cells to therapeutic agents. They also noticed a synergistic effect of inhibitors of EGFR and sorafenib over the activity of the RAF-MEK-ERK kinase cascade in EGFR-positive HCC cells [51]. The phase III SEARCH trial was a randomized, double-blind, placebo-controlled trial, which compared erlotinib and sorafenib with sorafenib monotherapy in patients with untreated, advanced HCC. The trial included 720 patients, and the inclusion criteria were unrespectable HCC, Child–Pugh liver function class A, life expectancy ≥12 weeks, and an ECOG-PS of 0 or 1. No significant difference in the median OS of the two groups was detected (9.5 vs. 8.5 months, HR = 0.929, *p* = 0.408). In addition, there was no difference in the TTP (3.2 vs. 4.0 months, HR = 1.135, *p* = 0.18). As a result, on the basis of this data analysis, the addition of erlotinib to sorafenib did not manage to improve survival in patients with advanced HCC [52].

Erlotinib has also been studied as part of other combination treatments. The efficacy of gemcitabine, oxaliplatin, and erlotinib (G + O + E) was evaluated in a phase II trial including 26 HCC patients. The response to treatment was evaluated with CT. Amongst the patients with HCC, 1 had a partial response and 10 had stable disease, thus a DCR of 42% at 24 weeks. The median PFS was 35 weeks and median OS 26 weeks. The difference observed between the PFS and OS was attributed mainly to cirrhosis-related deaths without disease progression [107].

#### 3.1.6. Foretinib

Foretinib is an oral MKI of MET, ROS, RON, AXL, TIE-2, and VEGFR2 (Figure 2F). A single-arm, phase I/II study was conducted in Asia in order to determine its safety and efficacy as a first-line treatment in patients with advanced HCC and Child–Pugh A liver disease. The maximum tolerated dose was 30 mg/day, and in this dose no major toxicity was demonstrated. Among the 35 patients evaluated for efficacy, the median TTP was 4.24 months (95% CI: 2.79–9.59). The DCR was 82.9% (95% CI: 66.4–93.4) and the median OS was 15.7 months [35].

As a result, foretinib seems to be efficient against advanced HCC, but randomized phase III trials are needed to prove this conclusion.

#### 3.1.7. Donafenib

Donafenib is a novel, oral, small-molecule MKI. In fact, donafenib is a modified form of sorafenib with a trideuterated N-methyl group (Figure 2G). After phase I studies showing a promising efficacy and safety profile, a randomized, open-label, parallel-controlled, phase II/III trial was conducted in a Chinese population. Finally, 659 patients with advanced HCC and Child–Pugh A, who had not received any prior systematic treatment, comprised the analysis population of the study. Among them, 328 were treated with donafenib and 331 with sorafenib, and 594 of them had HBV-related HCC. The ORR and the DCRs were 4.6% and 30.8%, respectively, in the donafenib group while 2.7% and 28.7%, respectively, in the sorafenib group. The median PFS for donafenib versus sorafenib was 3.7 vs. 3.6 months (HR = 0.909; 95% CI: 0.763–1.082; *p* = 0.0570), and the median TTP was 3.7 vs. 3.7 months (HR = 0.931; 95% CI: 0.777–1.117; *p* = 0.1029). However, a significant difference was observed in the OS, which was also evident in most subgroups. Specifically, the OS in donafenib arm versus sorafenib arm was 12.1 vs. 10.3 months, with an HR = 0.831 (95% CI: 0.699–0.988; *p* = 0.0245). The safety profile was in favor of donafenib too. There was significantly fewer drug-related grade ≥3 AEs with donafenib than sorafenib treatment (125 (38%) vs. 165 (50%); *p* = 0.0018). A significantly larger number of sorafenib-treated patients were led to dose interruption and reduction due to AEs (141 (42%) vs. 101 patients (30%), *p* = 0.0025), among which 84 (25%) and 120 (36%) were drug-related, respectively. The number of patients discontinuing treatment due to AEs was also higher in the sorafenib arm (34 (10%) vs. 42 (13%)). Regarding death-related AEs, there were six (2%) in the donafenib arm and 12 (4%) in the sorafenib arm [108].

Thus, donafenib could also be a part of the first-line treatment of advanced HCC, especially in selected populations.

Figure 2 shows the aforementioned multi-targeted TKI for the first-line treatment of HCC.
Figure 2Mechanism of action of multi-targeted tyrosine kinase inhibitor for first-line treatment of HCC. (**A**) Sorafenib; (**B**) lenvatinib; (**C**) sunitinib; (**D**) linifanib; (**E**) erlotinib; (**F**) foretinib; (**G**) donafenib.
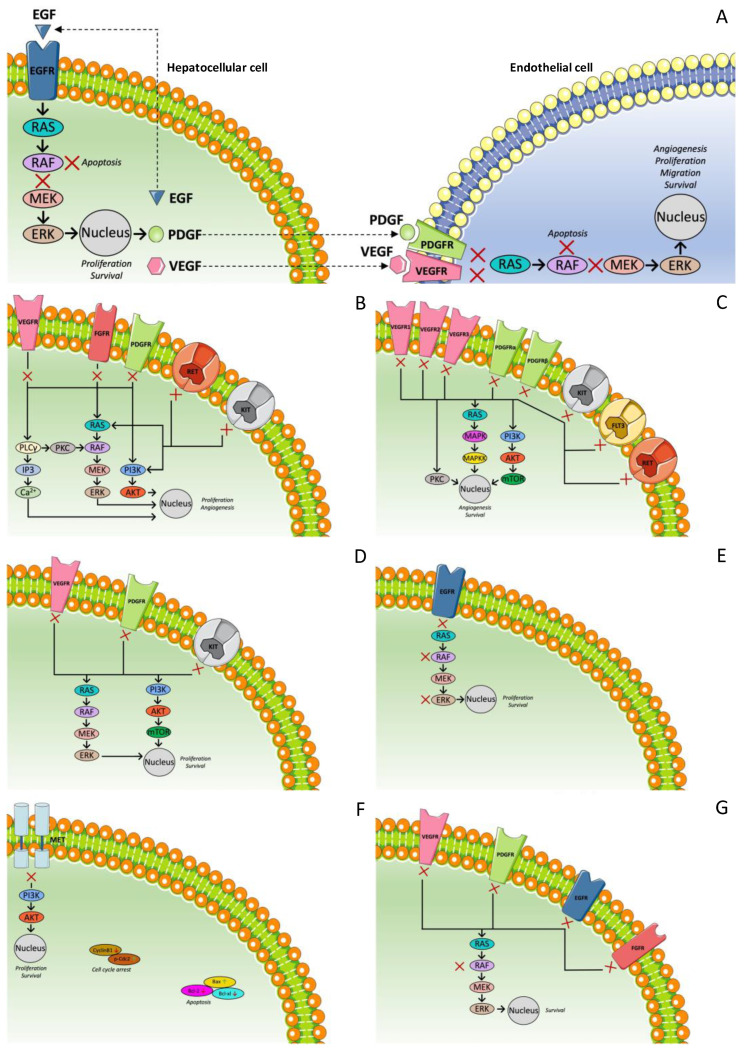


### 3.2. Multi-Targeted Tyrosine Kinase Inhibitors: Second-Line Treatment

#### 3.2.1. Regorafenib

Regorafenib is another oral MKI of VEGFR 1-3, TIE 2, PDGFR beta, FGFR, RET, KIT, RAF-1, and B-Raf (Figure 3A) [64,65,66]. Regorafenib has received approval as a monotherapy for the treatment of refractory colorectal cancer as well as for gastrointestinal (GI) stromal tumors [59,109]. As far as HCC is concerned, its efficacy and safety were first tested in a phase II study, and the results showed both anticancer activity and acceptable tolerability. Based on these results, the RESORCE trial was designed in order to investigate the benefit of regorafenib in patients with disease progression after the first-line treatment with sorafenib. Τhis phase III, double-blinded, placebo-controlled trial evaluated 573 patients with BCLC stage B or C and Child–Pugh A liver function who were tolerant to sorafenib and had demonstrated radiological progression during treatment with sorafenib and received its last dose within 10 weeks of randomization. Patients who had received other previous systemic treatment for HCC or had interrupted sorafenib because of toxicity were excluded. These patients were randomly assigned to regorafenib 160 mg once daily (for three weeks on and one week off) or placebo. The OS was set as the primary endpoint. According to the outcomes, regorafenib improves the median OS (10.6 vs. 7.8 months, HR = 0.63, *p* < 0.0001) and median PFS (3.1 vs. 1.5 months, HR = 0.46, *p* < 0.0001) and causes significantly higher rates of ORR (11% vs. 4%) [110].

Thus, this study met its primary endpoint and showed a significant prolongation in OS in patients with disease progression during first-line treatment with sorafenib, suggesting that the sequential use of two MKIs with partly overlapping targets provides a survival benefit in HCC [66].

#### 3.2.2. Cabozantinib

Cabozantinib is an oral TKI that suppresses MET, VEGFR-2, and RET (Figure 3B) [56]. A phase II randomized discontinuation trial (RDT) of 41 HCC patients with Child–Pugh A liver function who had previously received ≤1 systemic agent displayed a median OS of 11.5 months and a median PFS of 5.2 months [111]. Thus, further investigation in a phase III trial had been initiated. The CALESTIAL trial tested the efficacy of cabozantinib in patients with advanced HCC, Child–Pugh A, and an ECOG-PS of 0 or 1 who had previously received at least one but no more than two systematic treatments, one of which had been sorafenib. In the trial, 707 patients were randomized 2:1 to receive 60 mg of cabozantinib once daily or placebo. The trial met its primary endpoint after the second interim analysis, and the outcomes provided a statistically significant and clinically meaningful improvement in the median OS compared to placebo (10.2 vs. 8.0 months, HR = 0.76, 95% CI: 0.63–0.92, *p* = 0.0049). As far as the secondary endpoints are concerned, both the median PFS (5.2 vs. 1.9 months, HR = 0.44, *p* < 0.001) and ORR (4 vs. 0.4%, *p* = 0.0086) were significantly better [112]. Because of the fact that the study also included patients receiving third-line treatment, another post hoc analysis was conducted including the patients who had previously received only sorafenib too. There were 495 such patients. Regarding the duration of treatment with sorafenib, 136 were treated for less than 3 months, 141 for 3 to 6 months, and 217 for more than 6 months. Among them, 331 had received cabozantinib and 164 placebo. Both the OS and PFS were improved by cabozantinib in these patients regardless of the duration of the treatment with sorafenib. The median OS was 11.3 months in the cabozantinib arm versus 7.2 months in the placebo arm. There also seemed to be a better result for cabozantinib treatment in patients with a longer duration of previous treatment with sorafenib [113].

Thus, cabozantinib can constitute an option as a second- or third-line treatment in patients with advanced HCC.

#### 3.2.3. Tivantinib

Tivantinib selectively inhibits c-MET. MET (mesenchymal–epithelial transition factor) is a tyrosine kinase receptor encoded by the protooncogene c-MET. After binding to hepatocyte growth factor (HGF), the MET signaling pathway is activated. This pathway interferes in multiple cellular processes, such as differentiation and angiogenesis, and also controls cell invasion and metastasis (Figure 3C). Hence, dysregulated MET expression is involved in a variety of human cancers including HCC. In addition, MET over-expression is associated with poorer prognosis [114]. After the conduction of phase I and Ib trials, a randomized phase II trial, comparing the use of tivantinib vs. placebo as second-line treatments, assessed 71 patients. The analysis showed that the TTP was longer in patients treated with tivantinib than placebo (1.6 vs. 1.4 months, HR = 0.64, 90% CI: 0.43–0.94, *p* = 0.04). Specifically, in the subset of patients with an elevated expression of c-MET determined by immunohistochemistry, the median TTP showed a more notable improvement with tivantinib compared to placebo. (2.7 vs. 1.4 months, HR = 0.43, 95% CI: 0.19–0.97, *p* = 0.03). In addition, the median OS was also better with tivantinib (7.2 vs. 3.8 months, HR = 0.38, 95% CI: 0.18–0.81, *p* = 0.01). On the other hand, neutropenia and anemia were much more frequent and severe in the tivantinib group, and there were four deaths due to neutropenia in this group [115,116]. Therefore, a phase III, randomized, double-blind, placebo-controlled study was conducted. The study included 340 patients with unresectable and histologically confirmed HCC, an Eastern Cooperative Oncology Group performance status of 0–1, high MET expression (MET-high; staining intensity score ≥2 in ≥50% of tumor cells), Child–Pugh A cirrhosis, and radiographically confirmed disease progression after receiving sorafenib-containing systemic therapy. They were randomized 2:1 to tivantinib vs. placebo, while the cohort with a lower dose than the dose of the phase II trial was continued due to severe AEs in the other cohort. The median OS was similar in the tivantinib (8.4 months; 95% CI: 6.8–10.0) and placebo (9.1 months; 7.3–10.4) groups. No differences were observed regarding the median PFS, median TTP, and DCR. As a result, no effectiveness of tivantinib was proven [117]. In another randomized, double-blind, placebo-controlled, phase III study of tivantinib with similar inclusion criteria and dosage, including 195 patients, the median PFS was 2.8 months (95% CI: 2.7–2.9) in the tivantinib group and 2.3 months (95% CI: 1.5–2.8) in the placebo group, and the median OS was 10.3 months (95% CI: 8.1–11.6) in the tivantinib group and 8.5 months (95% CI: 6.2–11.4) in the placebo group. However, the results were not statistically important [118].

Thus, tivantinib has not been proven to be an effective second-line agent, but possibly concomitant inhibition of c-MET and VEGF could have better results [117].

#### 3.2.4. Axitinib

Axitinib is an oral, potent, selective inhibitor of VEGFR 1, 2, and 3 (Figure 3D). A single-arm phase II trial was conducted including patients with advanced HCC, Child–Pugh A/B7, and progressive disease after treatment with TKI/antiangiogenic drugs. Among the 26 patients evaluated for efficacy, the median OS for all patients was 7.1 months (95% CI: 6.3–13.7 months) and PFS 3.6 months (95% CI: 2.8–9.2 months). At 16 weeks, the tumor control rate was 42.3% (95% CI: 22.3–63.1), with 1 patient having a partial response and 10 having stable disease. Regarding toxicity, there were two patients with grade 4 thrombocytopenia, and eight patients (26.7%) discontinued treatment because of AEs, of which four cases were considered related to axitinib [119]. Nonetheless, a randomized phase II study of axitinib versus placebo plus best supportive care in second-line treatment of advanced HCC failed to prove its supremacy. Patients with locally advanced or metastatic HCC and Child–Pugh class A who progressed on or were intolerant to one prior antiangiogenic therapy were enrolled. Finally, 134 patients were included in the axitinib arm and 68 in the placebo. The median (95% CI) OS was 12.7 (10.2–14.9) months with axitinib/BSC and 9.7 (5.9–11.8) months with placebo/BSC, but the difference was not statistically important. On the other hand, the difference in the OS was even greater when patients intolerant to prior antiangiogenic treatment were excluded, and, in addition, the PFS and TTP were significantly longer in the axitinib arm [120]. Another phase II trial, studying the combination of TACE and axitinib as a first-line treatment of inoperable HCC, showed possibly favorable results. It included 50 patients with inoperable HCC who were potential candidates for TACE. The 2-year survival rate was 43.7%, and the 1-year survival rate was 72.0%. The median OS was 18.8 months (95% CI: 14.5–28.9 months), and the median TTP and PFS were 10.4 months (95% CI: 5.4–12.7 months) and 8.4 months (95% CI: 3.9–11.2 months), respectively. Regarding toxicity, bleeding was not common, while 24% of the patients developed hypertension of grade ≥3. The fact that the development of hypertension was an independent negative prognostic factor underlines the potential therapeutic importance of axitinib [121].

In conclusion, axitinib seems to have important antitumor activity in HCC, but further study with phase III clinical trials is needed.

#### 3.2.5. Anlotinib

Anlotinib is an orally administered TKI that targets VEGFR, FGFR, PDGFR, and c-kit (Figure 3E). A phase II clinical trial was conducted including two cohorts, one as a first-line and one as a second-line treatment. Cohort 1 included 26 patients without previous treatment with TKI, and cohort 2 included 24 patients who had received previous treatment. In cohort 1, the 12-week PFS rate was 80.8% (95% CI: 59.8–91.5%), and the 24-week PFS rate was 54.2% (95% CI: 32.4–71.7%). The median TTP was 5.9 months (95% CI: 4.8–6.9), and the median OS was 12.8 months (95% CI: 7.9–20.1). In cohort 2, the 12-week PFS rate was 72.5% (95% CI: 48.7–86.6%), and the 24-week PFS rate was 46.6% (95% CI: 24.4–66.2%). The median TTP and OS were 4.6 months (95% CI: 2.7–10.0) and 18.0 months (95% CI: 9.1–23.0), respectively. Additionally, the safety profile was favorable, while the median TTP of patients with a baseline plasma level of CXCL1 (C-X-C motif chemokine ligand 1) less than 7.6 ng/μL was significantly longer in both cohorts. Among the 25 patients with follow-up ≥2 scans at 12 weeks, the confirmed ORR was 24% (6/25) and DCR was 84% (21/25) [122]. Anlotinib has also been studied as first-line therapy in combination with penpulimab (Anti-PD-1) [123].

Thus, anlotinib seems to have satisfactory efficacy both as a first-line and second-line treatment for advanced HCC. Further studies are needed to prove this and to study the efficacy of combination therapies with anlotinib.

#### 3.2.6. Tepotinib

Tepotinib is an orally available, potent, and highly selective MET inhibitor (Figure 3F). In a phase Ib study, a dose of 500 mg per day was chosen, and a phase II trial was conducted. There were enrolled 49 patients with advanced HCC and Child–Pugh class A liver who had received at least 4 weeks’ treatment with sorafenib, which was discontinued due to intolerance or disease progression. The median TTP was 2.1 months (90% CI: 1.4–7.2), median PFS was 1.5 months (90% CI: 1.4–3.7), and median OS was 7.2 months (90% CI: 3.7–10.1). The DCR was 57.1% (90% CI: 44.4–69.2), as out of 49 patients, 24 had stable disease, 1 had a complete response, and 3 had a partial response. No major safety issues were reported. Patients with MET over-expression seemed to have a more favorable efficacy profile [124]. In another phase Ib/II study with Asian patients, tepotinib seemed to improve TTP in comparison with sorafenib without any overwhelming toxicity. Ninety patients with advanced HCC, Child–Pugh class A liver, and confirmed MET over-expression were randomly divided into two groups to receive either 500 mg/day tepotinib (n = 45) or sorafenib (n = 45) as a first-line treatment. The median independent review committee (IRC)-assessed TTP in the tepotinib arm was 2.9 months (90% CI: 2.7–5.3) versus 1.4 months (90% CI: 1.4–1.6) in the sorafenib arm (HR = 0.42, 90% CI: 0.26–0.70, *p* = 0.0043). The investigator-assessed TTP also favored tepotinib (median, 5.6 months; 90% CI: 3.0–7.6) versus sorafenib (median, 2.8 months; 90% CI: 1.5–2.8 months, HR = 0.45, 90% CI: 0.28–0.73, *p* = 0.0059). Similar results were found regarding the PFS, whereas no difference was found regarding the OS; 9.3 months and 8.6 months in the tepotinib and sorafenib arms, respectively (HR = 0.73, 90% CI: 0.43–1.21). Disease control was 50% (19/45) in the tepotinib arm and 21.6% (8/45) in the sorafenib arm [125].

Thus, tepotinib seems to be an efficient agent as both a first- and second-line treatment for advanced HCC, especially in patients with MET over-expression. Further phase III trials are needed to confirm these results.

Figure 3 describes the aforementioned multi-targeted TKI for second-line treatment of HCC.
Figure 3Mechanism of action of multi-targeted tyrosine kinase inhibitor for second-line treatment of HCC. (**A**) Regorafenib; (**B**) cabozantinib; (**C**) tivantinib; (**D**) axitinib; (**E**) anlotinib; (**F**) tepotinib.
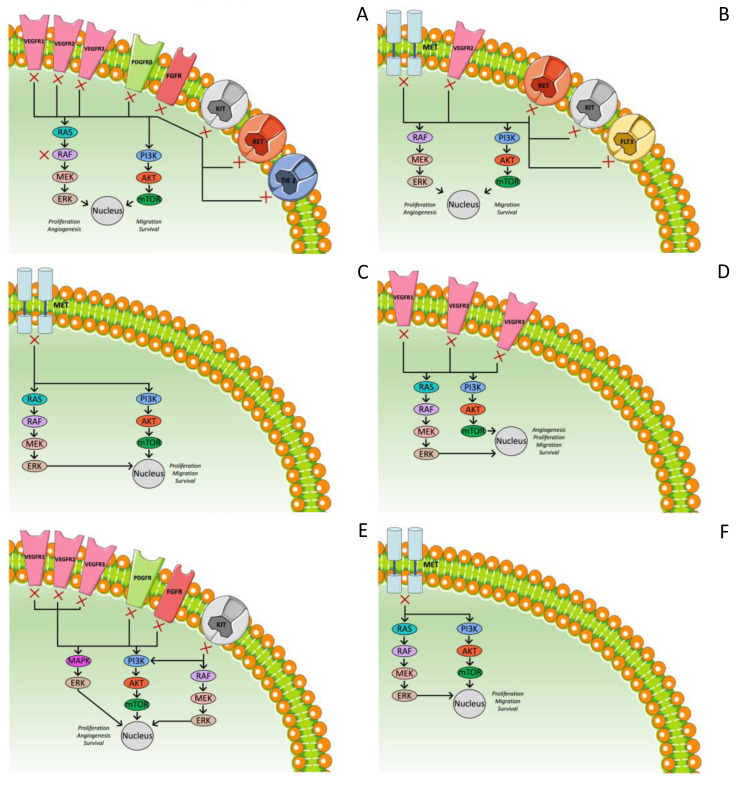


### 3.3. Anti-VEGF Therapies

#### 3.3.1. Ramucirumab

Ramucirumab is an injected agent. It is a human IgG1 monoclonal antibody, which targets the extracellular domain of VEGFR2, thereby blocking the binding of ligands and inhibiting the activation of the receptor-mediated pathway in endothelial cells (Figure 4A) [126]. A phase II trial showed the potential anticancer activity of ramucirumab as a first-line monotherapy [127], whereas the phase III REACH trial assessed the efficacy of ramucirumab compared to placebo as a second-line treatment in 565 patients with advanced HCC who failed or were intolerant to sorafenib. No superiority of ramucirumab in terms of the OS was noticed (9.2 vs. 7.6 months, HR = 0.87, *p* = 0.14), but in the secondary endpoints, the PFS (2.8 vs. 2.1 months, HR = 0.63, *p* < 0.001) and objective response (7% vs. <1%) favored the ramucirumab arm. Further analysis in subgroups showed that an elevated baseline AFP level was associated with ramucirumab efficacy. In detail, among patients with AFP ≥400 ng/mL, the median OS was 7.8 months for the ramucirumab arm vs. 4.2 months for the placebo arm (HR = 0.67, *p* = 0.006) [128].

Thus, based on this observation, the REACH-2 study, a randomized, double-blind, placebo-controlled, phase III study, was conducted in order to evaluate ramucirumab vs. placebo as second-line treatments in patients with AFP ≥400 ng/mL. The study enrolled 292 patients with BCLC stage B or C disease that was refractory or not amenable to locoregional therapy, with Child–Pugh class A liver disease and serum AFP concentrations of 400 ng/mL or higher, and who had received previous treatment only with sorafenib. The patients received intravenous ramucirumab (8 mg/kg) or placebo for 1 h every 14 days; 197 received ramucirumab and 95 placebo. The primary endpoint was the OS and it was reached as there was a statistically significant difference between the two arms; the OS was 8.5 months (95% CI: 7.0–10.6) in the ramucirumab arm vs. 7.3 months [5.4–9.1] in the placebo arm, with an HR = 0.710 (95% CI: 0.531–0.949), *p* = 0.0199. Additionally, the median PFS was significantly longer in the ramucirumab group than the placebo group (2.8 months (95% CI: 2.8–4.1) vs. 1.6 months [1.5–2.7]; HR = 0.452 (95% CI: 0.339–0.603), *p* < 0.0001, and the DCR was significantly higher in the ramucirumab group than in the placebo group: 118 (59.9%; 95% CI: 53.1–66.7) of 197 vs. 37 (38.9%; 95% CI: 29.1–48.8) of 95, *p* = 0.0006. When these results were adjusted to the levels of AFP, the difference in the OS remained significantly longer in the ramucirumab group. No extreme toxicity was noticed [129].

Ramucirumab was also studied in combination with emibetuzumab, a humanized IgG4 bivalent mAb that prevents human growth factor (HGF) from binding to the extracellular domain of MET and triggers MET receptor internalization and the degradation of total membrane MET expression; thus, it inhibits both ligand-dependent and ligand-independent activation of MET signaling. In a phase Ib/II study, after the dose of phase II was determined, 97 patients with four different solid tumors were enrolled. Among them, there were 45 with HCC with a Child–Pugh A score, while 18 patients (40%) had AFP >400 ng/mL. Of these, 37 (82%) had received previous sorafenib treatment, 3 (4%) had received prior therapies other than sorafenib, and 5 (11%) had received this combination as first-line treatment. The ORR was 6.7%, the DCR 60% (95% CI: 0.4–0.7), and the median PFS 5.4 months (95% CI: 1.6–8.1) [130].

Consequently, ramucirumab has shown satisfactory efficacy against advanced HCC as a second-line treatment in a selected population with high AFP levels, while it may also have promising antitumor activity as part of a combination therapy.

#### 3.3.2. Bevacizumab

Bevacizumab is a humanized monoclonal antibody that binds VEGF-A (Figure 4B). It has been studied for years as a possible treatment for advanced HCC. Several phase II trials have examined the safety and efficacy of bevacizumab on HCC either as a monotherapy or as a combination treatment. In a phase II trial enrolling 46 patients, of whom two-thirds had also received another treatment previously, the median PFS time was 6.9 months (95% CI, 6.5 to 9.1 months), and the OS rate was 53% at 1 year, 28% at 2 years, and 23% at 3 years [131]. In another phase II trial, among 38 patients the DCR at 16 weeks was 42% (95% CI: 27%–57%). The median PFS was 3 months (95% CI, 2–4 months), and the median OS duration was 8 months (95% CI: 4–9 months) [132]. In both studies, the patients enrolled had Child–Pugh A or compensated B. No major toxicity was noticed; the main AEs included hypertension and GI bleeding. The combination of bevacizumab with gemcitabine and oxaliplatin showed possible anticancerous activity in a phase II study [133], as did the combination of bevacizumab with capecitabine and oxaliplatin as a first-line treatment in a phase II study, which also included patients with Child–Pugh B [134]. The median OS was 9.6 months (95% CI, 8.0 months to not available) and 9.8 months (95% CI: 5.2–12.1 months), and the median PFS was 5.3 months (95% CI: 3.7–8.7 months) and 6.8 months (95% CI: 3.4–9.1 months), respectively. Bevacizumab was also studied as a first-line treatment in a combination therapy with oral capecitabine demonstrating antitumor activity. In a phase II study with 45 patients, the median PFS was 2.7 months (95% CI: 1.5–4.1 months), and the median OS was 5.9 months (95% CI: 4.1–9.7 months) [135].

Bevacizumab has also been studied in combination with erlotinib. In two phase II studies using this combination mainly as a first-line treatment, a promising efficacy was found [136,137]. In the first, among 40 patients the median PFS was 39 weeks/9.0 months (95% CI: 26–45 weeks), and the median OS was 68 weeks/15.65 months (95% CI: 48–78 weeks) [136], while in the second, among 59 patients the median OS was 13.7 months (95% CI: 9.6–19.7), and the median PFS 7.2 months (95% CI: 5.6–8.3) [137]. On the other hand, two other phase II trials failed to show the efficacy of the combination [138,139]. Yau et al. found a very poor response among 10 patients with progression of disease under prior treatment with sorafenib [138], while Philip et al. found no improved efficacy in the combination of bevacizumab and erlotinib. Among 27 patients, of whom only 1 had received previous systematic treatment, the median time to disease progression was 3.0 months (95% CI: 1.8–7.1) and median survival time 9.5 months (95% CI: 7.1–17.1) [139].

Nonetheless, Thomas et al. compared this combination with sorafenib in a randomized phase II study including 90 patients with 45 in each arm. This study also included patients with Child–Pugh class B cirrhosis. The combination therapy was proved to be neither better nor worse than sorafenib alone. There was not a statistically significant difference in the OS, response rate, or event-free survival. On the other hand, the sorafenib-treated arm displayed more AEs related to treatment and more cases of discontinuation of therapy; thus, the combination seemed to have a more favorable safety and tolerability profile [140].

Bevacizumab has also been studied as a combination therapy with TACE. Despite the fact that a phase II trial (25 patients) showed potential efficacy without reaching its primary endpoint of time to tumor progression [141] and a phase II study comparing TACE plus bevacizumab with TACE alone (15 patients on each arm) showed no significant difference in the OS but significant difference in the 16w-PFS [142], another phase II randomized controlled double-blind trial of TACE in combination with biweekly intravenous administration of bevacizumab or a placebo (20 patients on each arm) showed worse OS for the combination group, no difference in radiological response, and a worse safety profile for the group of TACE plus bevacizumab [143].

However, a global, open-label, phase III trial comparing bevacizumab plus atezolizumab created a new era in the treatment of advanced HCC. Atezolizumab is a selective programmed death 1 (PD-1) inhibitor that inhibits interaction with receptors PD-1 and B7-1 reversing T-cell suppression [144]. After promising results in a phase Ib study comparing atezolizumab and bevacizumab with atezolizumab alone [144], a phase III trial was conducted comparing bevacizumab and atezolizumab with sorafenib as a first-line treatment. In this trial, 501 patients with unresectable HCC and Child–Pugh A were randomized to two groups: 336 received 1200 mg of atezolizumab plus 15 mg per kilogram of body weight of bevacizumab intravenously every 3 weeks, while 165 patients received 400 mg of sorafenib orally twice daily. After a median follow-up of 8.6 months, the hazard ratio for death with atezolizumab–bevacizumab in comparison with sorafenib was 0.58 (95% CI: 0.42–0.79; *p* < 0.001). The OS at 12 months was 67.2% (95% CI: 61.3–73.1) with atezolizumab–bevacizumab and 54.6% (95% CI: 45.2–64.0) with sorafenib. The median PFS was 6.8 months (95% CI: 5.7–8.3) vs. 4.3 months (95% CI: 4.0–5.6), respectively. The DCR was 73.6% vs. 55.3%. There was also a significant difference in confirmed ORR using both RECIST 1.1 and HCC–specific mRECIST criteria. As far as the combination of bevacizumab plus atezolizumab is concerned, safety issues were manageable. The most common grade 3 or 4 AE was hypertension. Discontinuation of therapy occurred in 15.5% (7% discontinued both components) in the atezolizumab–bevacizumab group and 10.3% in the sorafenib group [145].

Thus, bevacizumab in combination with atezolizumab has been found superior to sorafenib as a first-line treatment of advanced HCC, and it is now considered the new standard of care.

#### 3.3.3. Apatinib

Apatinib is a highly selective and potent TKI that blocks angiogenesis by targeting VEGFR 2 (Figure 4C). In a randomized, double-blind, placebo-controlled, multicenter, phase III trial, apatinib was compared to placebo as a second-line treatment of advanced HCC. Among the 393 patients participating in the trial who were eligible for further analysis, 261 received apatinib and 132 placebo. All of them had a Child–Pugh score of 7 or lower, while 41% had received sorafenib as first-line treatment, and about one-fifth of participants had received more than one systematic therapy prior to apatinib. The median OS was 8.7 months (95% CI: 7.5–9.8) in the apatinib group versus 6.8 months (5.7–9.1) in the placebo group. The median PFS was 4.5 months (95% CI: 3.9–4.7) in the apatinib group and 1.9 months (1.9–2.0) in the placebo group (HR = 0.471 (95% CI: 0.369–0.601), *p* < 0.0001). Objective response and disease control were also higher in the apatinib arm. After adjusting for poststudy treatment, the efficacy of apatinib was even higher. Apatinib also seemed to have an even more favorable effect on patients aged 65 years or younger, patients with AFP greater than or equal to 200 µg/L, patients with no previous sorafenib treatment, and patients with only one previous systemic therapy. The 95% CIs for the 12-month PFS and 12-month OS estimates overlapped between the apatinib and placebo groups. Due to related AEs, 32 patients in the sorafenib group (12%) vs. 0 (0%) in the placebo group discontinued treatment. In patients treated with apatinib, the most common grade 3 or 4 treatment-related AE was hypertension (71 (28%) patients). In general, no major toxicity attributed to treatment with apatinib was noticed [146].

In addition, after the encouraging results of a phase I study [147], a non-randomized, open-label, phase II trial assessed the efficacy of apatinib as part of a combination treatment with camrelizumab (a high-affinity, humanized IgG4-κ PD-1 mAb). The trial enrolled 190 patients with advanced HCC and Child–Pugh A. Among them, 70 received the combination as first-line treatment and 120 as second-line. The ORR was 34.3% (24/70; 95% CI: 23.3–46.6) in the first-line and 22.5% (27/120; 95% CI: 15.4–31.0) in the second-line cohort per the independent review committee, RECIST v1.1. The 12-month survival rate was 74.7% (95% CI: 62.5–83.5) and 68.2% (95% CI: 59.0–75.7), respectively. The median PFS was 5.7 months (95% CI: 5.4–7.4) and 5.5 months (95% CI: 3.7–5.6), respectively. Disease control was reported in 54 (77.1%; 95% CI: 65.6–86.3) of the 70 patients in the first-line cohort and in 91 (75.8%; 95% CI: 67.2–83.2) of the 120 patients in the second-line cohort. Regarding safety issues, 23 (12.1%) of 190 patients discontinued camrelizumab combined with apatinib because of treatment related AEs. Two deaths were reported (1.1%) related to the drugs in the study. The most common AE was hypertension. Consequently, this combination showed very promising results [148]. A randomized, open-label, international, multicenter, phase III clinical study comparing this combination with sorafenib as a first-line treatment of advanced HCC is being conducted (NCT03764293).

Apatinib has also demonstrated favorable results in combination with TACE. In a single-center randomized controlled trial, 44 patients with moderate and advanced HCCs and Child–Pugh A or B were randomized 1:1 to receive either TACE or TACE plus apatinib. Apatinib was stopped 4 days before TACE and recontinued 4 days after. The median PFS was 6.0 months in group A and 12.5 months in group B, and the difference was statistically significant (*p* < 0.05). However, the fall in AFP after 3 months of treatment was not statistically significant between the two groups. The difference in ORR at 9 and 12 months was statistically significant, while that of 3 and 6 months not [149].

Thus, apatinib has been proven to be an effective second-line treatment, while it could also bring a new era in the treatment of HCC as part of combination therapies.

Figure 4 describes the aforementioned anti-VEGF therapies for the treatment of HCC.
Figure 4Mechanism of action of anti-VEGF therapies for the treatment of HCC. (**A**) Ramucirumab; (**B**) bevacizumab; (**C**) apatinib.
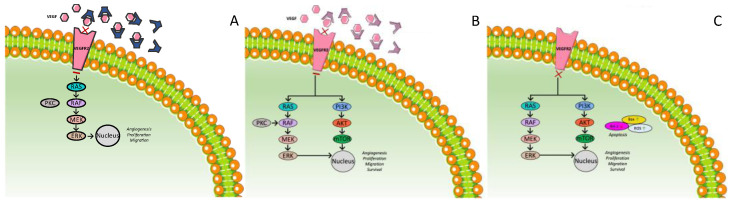


### 3.4. TGF-β Receptor Inhibitor

TGF-β signaling is a therapeutic target against HCC [150]. Galunisertib (LY2157299 monohydrate) is an oral small-molecule inhibitor of the TGF-β receptor I kinase (Figure 5) [151]. In a phase II trial studying galunisertib as a second-line treatment, 149 patients with advanced HCC and Child–Pugh class A or B7 were enrolled, of whom 109 had AFP ≥ 1.5 × upper limit of normal (ULN) (Part A) and 40 had AFP < 1.5 × ULN (Part B); 37 patients of group A received 80 mg galunisertib twice daily on 14 consecutive days followed by 14 days rest in a 28-day cycle, while the rest of the patients in both groups received 150 mg, and 123 out of 149 had previously received sorafenib. The median TTP was 2.7 months (95% CI: 1.5–2.9) in Part A and 4.2 months (95% CI: 1.7–5.5) in Part B. The median OS was 7.3 months (95% CI: 4.9–10.5) in Part A and 16.8 months (95% CI: 10.5–24.4) in Part B. The OS for Part A 150 mg BID vs. Part B 150 mg BID was significantly longer (HR = 2.1, 95% CI: 1.3–3.3), and the TTP for the AFP responders (a reduction of more than 20% from baseline within the first six cycles of treatment) was also significantly longer (4.3 (2.9–15.2) vs. 1.5 (1.4–2.8)) in comparison with non-responders. The safety profile was very favorable [152].

Additionally, the combination of galunisertib and sorafenib was studied as a first-line treatment in a phase II study in which 47 patients were enrolled, of whom 3 received galunisertib at a dose of 80 mg and 44 at a dose of 150 mg. Among these 44 patients, the TTP was 4.1 months (95% CI: 2.8, 6.5) and OS 18.8 months (95% CI: 14.8, 24.8). AFP and TGF-β1 seemed to have prognostic value. Apart from this, the TGF-β1 responders (decrease of >20% from baseline) vs. the non-responders had a longer OS (22.8 vs. 12.0 months, *p* = 0.038) [153].

In conclusion, the TGF-β pathway could also offer therapeutic alternatives for advanced HCC. Galunisertib is the one most studied to date, with favorable effectiveness, but further phase III trials are needed.

### 3.5. Immunotherapy

T-cells are responsible for entering the tumor microenvironment, detecting tumor-specific antigen–major histocompatibility complexes, and being activated after binding them to their T-cell antigen receptor [154]. In addition, in order for the activation to be successful, an additional signal is required. This co-stimulating signal is the engagement of CD28 on the T-cell surface with B7 molecules on the antigen-presenting cell (APC) [155]. T-cell activation, however, induces not only T-cell proliferation and differentiation but also an inhibitory pathway that could lead to the termination of T-cell responses. In detail, cytotoxic T-lymphocyte protein-4 (CTLA-4) is a T-cell membrane glycoprotein, which, as it is homologous to CD28, binds B7 molecules too but with greater affinity [156]. CTLA-4, in contrast with CD28, leads to the downregulation of T-cell responses [157]. Hence, the intensity of cytotoxic T-cell activation is determined by a balance between stimulatory and inhibitory signal pathways. Apart from CTLA-4, programmed cell death-1 (PD-1) has been recognized as an additional immune checkpoint. PD-1 is a type I transmembrane protein, related to CD28 and CTLA-4, which is expressed by activated T-cells, B cells, and myeloid cells. PD-1 has two ligands, PD-L1 (B7-H1) and PD-L2 (B7-DC), expressed by APC, which bind PD-1 and dephosphorylate TCR. The pathway of PD-1 and its ligands also interferes in the inhibitory pathways, which leads to the downregulation of T-cell responses [154,157]. Immune checkpoint treatment, obtained with monoclonal antibodies (mAbs), aim to amplify antitumor immune response via targeting molecules involved in the regulation of T-cells and suppressing the inhibitory pathways that prevent effective antitumor T-cell responses [158]. As a result, it appears to be a promising novel treatment strategy in the field of cancer.

Immune checkpoint therapy has already achieved some encouraging results in a variety of cancer types, such as metastatic melanoma, advanced renal cell carcinoma, bladder cancer, and head and neck cancer, with metastatic melanoma being the most representative among them, as ipilimumab, pembrolizumab, and nivolumab have already received FDA approval for its treatment [159,160,161,162,163,164,165]. As far as HCC is concerned, immunotherapy seems to be a promising strategy [166,167]. The immune system seems to be a key regulator of cancer development, playing a pivotal role against viral hepatitis as well as in the progression of chronic inflammation, with both of these contributing to tumor development [168]. Therefore, the efficacy of immune checkpoint inhibitors has been explored in clinical trials, providing some promising results.

#### 3.5.1. Nivolumab

Nivolumab is a fully human immunoglobulin G4 monoclonal antibody that targets PD-1 and thus restores antitumor T-cell activity (Figure 6A). CheckMate 040 was the first trial trying to evaluate the safety and efficacy of nivolumab in patients with advanced HCC with or without chronic viral hepatitis (HBV or HCV). It was a phase I/II trial, which included a dose-escalation and a dose-expansion phase. The eligibility criteria included disease progression or intolerability to first-line treatment, Child–Pugh liver function A or B7, and an ECOG-PS of 1 or less. Patients who had previously received another immune checkpoint inhibitor were excluded. In addition, for patients with HBV, the viral load was required to be less than 100 IU/mL and they also should have received effective antiviral therapy. Antiviral therapy was not a prerequisite for patients with HCV. More specifically, 48 patients were included in the dose-escalation phase, and 214, 68% of whom had previously been treated with sorafenib, in the dose-expansion phase. Safety and tolerability were set as primary endpoints for the escalation phase and ORR for the expansion phase. In the dose-escalation phase, patients received 0.1 to 10 mg/kg IV every two weeks. According to the analysis, the RR was 15%, DCR was 58%, median TTP was 3.4 months, and median OS was 15 months, whereas the median duration of response was 17 months. Grade 3/4 treatment-related adverse effects occurred in 25% of the patients. The dose of 3 mg/dl every 2 weeks was selected for the dose-expansion phase. The results of the analysis showed that the RR was 20% and the median duration of response 9.9 months. The DCR was 64%, median TTP was 4.1 months, and the 9-month OS rate was 74%. In addition, an update analysis of CheckMate 040 demonstrated a significant prolongation of the median survival (sorafenib-naive patients, 28.6 months, sorafenib-experienced patients, 15.6 months) [169]. The manageable toxicity profile of the drug, as well as the favorable responses that were noticed, led to the FDA approval of the agent.

Among the patients included in CheckMate 040, 49 had Child–Pugh B, 8 of whom (16%) were infected with HBV and 21 (43%) with HCV. In addition, 25 patients were sorafenib naive, while 24 were pre-treated with sorafenib. The ORR and DCR were 12% (95% CI: 5–25) and 55% (95% CI: 40–69), respectively. The median OS for all Child–Pugh B patients was 7.6 months (95% CI: 4.4–10.5). The median OS (95% CI) for sorafenib-naive and sorafenib-treated patients was 9.8 months (3.7–14.3) and 7.4 months (2.3–12.1), respectively. The median PFS (95% CI) for all Child–Pugh B patients was 2.7 months (1.6–4.0), and the median PFS for sorafenib-naive and sorafenib-treated patients was 3.4 months (1.6–4.1) and 2.2 months (1.4–4.2), respectively. Regarding the safety profile, no major differences were noted in comparison with Child–Pugh A patients. Two patients only (4%) discontinued treatment due to treatment-related AEs, while toxicity was also manageable in Child–Pugh B patients [170].

In the same clinical trial, the combination of nivolumab plus ipilimumab was also assessed. Ipilimumab is a cytotoxic T-lymphocyte–associated protein 4 (CTLA-4) immune checkpoint inhibitor. The trial included 148 patients with advanced HCC previously treated with sorafenib, who were randomized into three dosing arms. Arm A was treated with nivolumab 1 mg/kg plus ipilimumab 3 mg/kg every 3 weeks (4 doses) followed by nivolumab 240 mg intravenously every 2 weeks. Arm B was treated with nivolumab 3 mg/kg plus ipilimumab 1 mg/kg every 3 weeks (4 doses) followed by nivolumab 240 mg intravenously every 2 weeks. Arm C was treated with nivolumab 3 mg/kg every 2 weeks plus ipilimumab 1 mg/kg every 6 weeks. In arms A, B, and C, the investigator-assessed ORR was 32%, 27%, and 29%. The median OS was 22.8 months (95% CI: 9.4–not reached) in arm A vs. 12.5 months (95% CI: 7.6–16.4) in arm B and 12.7 months (95% CI: 7.4–33.0) in arm C. There were no major safety issues, although patients of arm A had a higher frequency of treatment-related AEs. Nine patients (18%) in arm A, three patients (6%) in arm B, and one patient (2%) in arm C discontinued treatment due to treatment-related AEs. As a result, the arm A dosage received approval in the US as second-line treatment of advanced HCC [171].

In a randomized, multicenter, open-label, phase III trial (CheckMate 459), nivolumab versus sorafenib was assessed as first-line treatments of advanced HCC. This trial included 743 patients with Child–Pugh A who were randomized to receive either nivolumab (371) or sorafenib (372). Nivolumab showed promising efficacy with an ORR of 15% versus 7% with sorafenib and a complete RR of 4% versus 1%, respectively. However, there was no significant difference in OS between the two groups. The median OS was 16.4 months with nivolumab and 14.7 months with sorafenib, while the PFS was 3.7 versus 3.8, respectively. Regarding the safety profile, the proportion of patients with grade 3–4 treatment-related AEs and any-grade treatment-related AEs leading to discontinuation was lower with nivolumab than with sorafenib. In fact, 22% of patients in the nivolumab arm had a grade 3–4 treatment-related AE and 4% discontinued treatment because of it, while in the sorafenib arm these were 49% and 8%, respectively [172].

#### 3.5.2. Pembrolizumab

Pembrolizumab is another anti-PD-1 monoclonal antibody (Figure 6A). In a non-randomized phase II trial, pembrolizumab was assessed as a second-line treatment of advanced HCC, after treatment with sorafenib. In this trial, 104 patients, 98 with Child–Pugh A and 6 with B, received treatment with pembrolizumab, 200 mg IV every 3 weeks. All of them were previously treated with sorafenib and none of them was pre-treated with another anti-PD agent. The ORR was 17%, 95% CI: 11–26 (18/104) and DCR was 62%, 95% CI: 52–71 (64/104). The median TTP was 4.9 months (95% CI: 3.9–8.0) and median PFS was 4.9 months (95% CI: 3.4–7.2). The safety profile was manageable. Treatment-related events of grade 3 or worse severity were reported in 27 (26%) participants and immune-mediated events occurred in 15 (14%) participants, while 5 (5%) discontinued the study treatment due to an AE [173].

Similar results were also shown by a randomized, double-blind, phase III trial including 413 patients, 410 with Child–Pugh A and 3 with B, who had previously received sorafenib and no other treatment with an anti-PD agent. A total of 278 patients received pembrolizumab and 135 placebo plus best supportive care. The ORR was 18.3% (95% CI: 14.0–23.4%) for pembrolizumab and 4.4% (95% CI: 1.6%–9.4%) for the placebo. The DCRs were 62.2% (95% CI: 56.2%–68.0%) and 53.3% (95% CI: 44.6%–62.0%), respectively. However, there was no significant difference in the median OS and PFS. The median OS was 13.9 months (95% CI: 11.6–16.0 months) in the pembrolizumab group and 10.6 months (95% CI: 8.3–13.5 months) in the placebo group and median PFS 3.0 months (95% CI: 2.8–4.1 months) and 2.8 months (95% CI: 1.6–3.0 months), respectively. The median time to progression was 3.8 months (range, 2.8–4.4 months) in the pembrolizumab group and 2.8 months (range, 1.6–2.9 months) for the placebo. Regarding the safety profile, 48 patients (17.2%) versus 12 (9.0%) in the placebo group discontinued treatment due to an AE, while 84 patients (30.1%) and 21 (15.7%) in the placebo arm interrupted treatment due to an AE. As far as immune-mediated AEs are concerned, they occurred in 51 patients (18.3%) in the pembrolizumab group and 11 (8.2%) in the placebo. One patient in the pembrolizumab arm died due to a treatment-related AE [174]. In addition, there was found to be no significant difference between the two groups in health-related quality of life evaluated with the European Organization for Research and Treatment of Cancer Core Quality of Life Questionnaire and its HCC supplement [175].

On the basis of this evidence, pembrolizumab monotherapy was approved as a second-line treatment of advanced HCC by the FDA but not by the European authorities because of the lack of benefit in OS and PFS.

#### 3.5.3. Avelumab

Avelumab is a fully human mAb targeting programmed death ligand 1 (PD-L1) antibody (Figure 6B) with known efficacy in Merkel cell carcinoma, urothelial carcinoma, and renal cell carcinoma. In a phase II study, the efficacy and safety of avelumab as a second-line treatment of advanced HCC was assessed. Thirty patients with Child–Pugh A, previously treated with sorafenib, who had not received another anti-PD factor, were treated with intravenous avelumab 10 mg/kg every 2 weeks. Of these, 26 had HBV-related HCC. The ORR was 10.0% (3 partial responses) and DCR was 73.3% (3 had partial responses and 19 had stable disease). The median PFS was 3.5 months (95% CI: 2.0–5.1) and median OS was 14.2 months (95% CI: 9.5–18.9). The avelumab response did not depend on PD-L1 and PD-1 expression, but a better response was associated with longer previous sorafenib treatment. The safety profile was favorable. Seven patients had grade 3 AEs, one had immune-related AEs due to treatment, and two patients discontinued treatment due to severe AEs (spontaneous bacterial peritonitis and rupture of HCC). There was no viral flare of HBV or HCV [176].

Thus, avelumab could be another therapeutic option for advanced HCC, although further evidence is needed to confirm it.

#### 3.5.4. Tremelimumab plus Durvalumab

The combination of tremelimumab, an anticytotoxic T-lymphocyte-associated antigen-4 (anti-CTLA-4) monoclonal antibody (Figure 6C), and durvalumab, an anti-PD-L1 monoclonal antibody (Figure 6B), has also been investigated in the treatment of advanced HCC. In a randomized phase I/II study, after the safety of the combination was evaluated, 207 patients with advanced HCC with no previous immunotherapy, who were intolerant to or progressed on sorafenib treatment or refused treatment with sorafenib, were randomized into four arms: 75 patients after a single dose of tremelimumab (300 mg IV, cycle 1) combined with durvalumab (1500 mg IV) continued receiving durvalumab (1500 mg IV) once every 4 weeks (T300 + D); 104 patients received durvalumab monotherapy (1500 mg IV once every 4 weeks); 69 patients received tremelimumab monotherapy (750 mg IV every 4 weeks, and after 7 doses once every 12 weeks); and 84 patients were treated with tremelimumab, 75 mg IV, once every 4 weeks for four cycles, and durvalumab, 1500 mg IV once every 4 weeks, (T75 + D). Tremelimumab monotherapy had the most treatment-related AEs (24.6%), while T300 + D, durvalumab, and T75 + D had 17.6%, 10.9%, and 14.6%, respectively. No significant difference between the four arms was noticed regarding discontinuation of treatment because of treatment-related AEs: 10.8%, 7.9%, 13.0%, and 6.1% for T300 + D, durvalumab, tremelimumab, and T75 + D, respectively. Additionally, regarding deaths possibly related to treatment, there was one patient in the T300 + D arm (pneumonia), three in the durvalumab arm (pneumonitis, abnormal hepatic function, and hepatic failure), and one in the T75 + D arm (hepatic failure). The T300 + D arm had the highest ORR (24.0%; 95% CI: 14.9–35.3) but also the highest median OS (18.73, 95% CI: 10.78–27.27 months). The median OS was 15.11 months (95% CI: 11.33–20.50), 13.57 (95% CI: 8.74–17.64), and 11.30 (95% CI: 8.38 –14.95) with tremelimumab, durvalumab, and T75 + D, respectively. The median PFS was 2.17 months (95% CI: 1.91–5.42) with T300 + D, 2.07 (95% CI: 1.84–2.83) with durvalumab, 2.69 (95% CI: 1.87–5.29) with tremelimumab, and 1.87 (95% CI: 1.77–2.53) with T75 + D [177].

Thus, the combination of T300 + D seems to be effective and safe as a second-line treatment of advanced HCC, but further studies are needed to establish the best dosage as well as the possible use of durvalumab monotherapy as another therapeutic option.

#### 3.5.5. Sintilimab

The combination of sintilimab, a selective anti-PD-1 antibody (Figure 6A), plus IBI305, a bevacizumab biosimilar, has been assessed as a first-line treatment of advanced HCC in a randomized, open-label, phase II/III study (ORIENT-32). After a manageable safety profile among 24 patients in the phase II part of the study, 595 patients were randomized to receive either sintilimab plus IBI305 (n = 380) or sorafenib (n = 191). Of these, the percentage with HBV-related HCC was 94%. The combination of sintilimab plus IBI305 proved to be more effective as a first-line treatment of HBV-related HCC in comparison with sorafenib. There was a significant difference in favor of the combination therapy regarding the ORR, DCR, and OS. In addition, the median PFS was 4.6 months (95% CI: 4.1–5.7) in the combination arm and 2.8 months (95% CI: 2.7–3.2) in the sorafenib arm, with a stratified HR = 0.56 (95% CI: 0.46–0.70; *p* < 0.0001). The time to deterioration of global health status was also significantly longer in the sintilimab plus IBI305 arm. There were not major safety issues. Treatment discontinuation due to a treatment-emergent AE occurred in 52 (14%) patients in the sintilimab–IBI305 group and 11 (6%) patients in the sorafenib group. Among them, there were seven [2%] upper GI bleedings in the sintilimab–IBI305 group vs. none in the sorafenib group. Treatment-related deaths occurred in ten (3%) and in six (3%) patients, respectively. The most frequent grade 3 or worse treatment-emergent AE was hypertension in both groups, but with a higher prevalence (15% vs. 6%) in the combination therapy group [178].

Thus, sintilimab plus IBI305, a bevacizumab biosimilar, outperformed sorafenib as a first-line treatment in HBV-related advanced HCC and could offer an alternative treatment option, especially in this specific population.

Figure 6 shows the aforementioned immunotherapies for the treatment of HCC.
Figure 6Mechanism of action of anti-VEGF therapies for the treatment of HCC. (**A**) Nivolumab, pembrolizumab, and sintilimab; (**B**) avelumab and durvalumab; (**C**) tremelimumab.
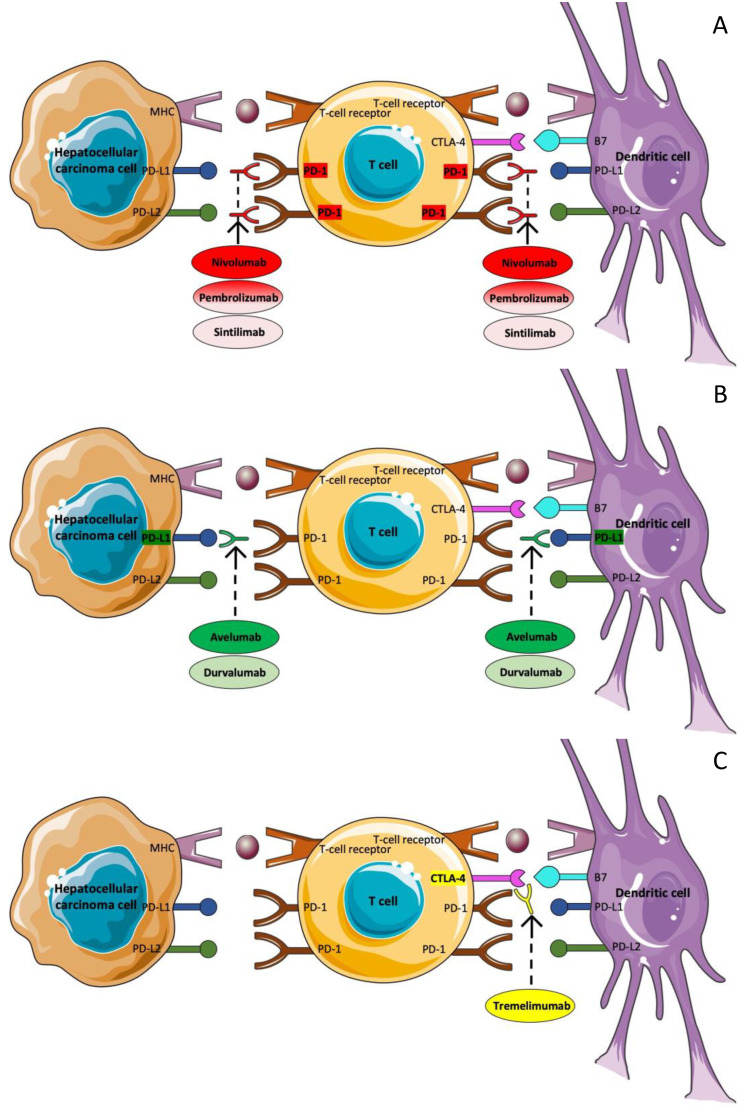


### 3.6. CAR-T Cell Therapy

Chimeric antigen receptor T-cells (CAR-T) are directed against specific tumor-associated antigens and consist of four components: the external single-chain variable fragment (scFv) domain, the IgG1 hinge−CH2−CH3 Fc domain that connects the scFv, the transmembrane part, and the intracellular part. The last consists of the CD3ζ signaling domain and other costimulatory domains specific for each CAR-T generation (Figure 7). CAR-T cell therapy has been approved for the treatment of hematological malignancies. There are also being conducted several studies examining the efficacy and safety of CAR-T therapy in HCC [179].

Endothelial progenitor cells (EPCs) play a critical role in neoangiogenesis, and CD133 is highly expressed in EPCs. In a single-arm, open-label, phase II trial, 21 patients were enrolled to receive CD133-directed CAR-T cells. Among them, 12 had Child–Pugh B, 18 had stage IV disease, and all of them had BCLC stage C and had previously been treated with at least one systematic therapy (16 had received sorafenib). The patients received 0.5 × 106 to 2 × 106 autologous CAR-T cells per kilogram of body weight, and if there was not any intolerable toxicity or tumor progression, they could receive another cell treatment cycle at least 4 weeks afterwards. The safety profile was quite manageable, with most AEs resolving after a few weeks. The DCR was 71.4% (1 patient with partial response and 14 with stable disease). The median PFS was 6.8 months (95% CI: 4.3–8.4 months), and the median OS was 12 months (95% CI: 9.3–15.3 months). After treatment, a decrease in the number of EPCs was noticed [180].

This is the first phase II clinical trial showing that CAR-T cell therapy could really become part of an advanced HCC treatment with hopeful efficacy.

Table 1 and Table 2 summarize the phase III trials that evaluated targeted therapies as first- and second-line treatments against HCC.

## 4. Discussion

Despite the progress and the availability of more therapeutic options, the overall prognosis in HCC remains mediocre. In the case of sorafenib, the results obtained in HCC led to an improved understanding of the disease and an advanced survival for patients. Sorafenib was the first systematic TKI to gain approval, and this could be considered as the first step on the way toward disease curation [82,83]. Since then, however, much progress has been made in this direction.

Although, at first, many results were inferior compared to sorafenib [67,101], the results of the IMBRAVE150 trial were a revolution in the first-line treatment of advanced HCC, as this combination has been established as the standard of care for improving the OS of HCC patients [145]. Indeed, the progression of immunotherapy seems to offer a lot of reliable first- and second-line treatments either as a monotherapy or as part of combination therapies [145,172,174,178]. Meanwhile, despite some initial disappointments, the antiangiogenic therapies seem able to participate crucially in the treatment of HCC. Apart from bevacizumab, ramucirumab and apatinib are used as second-line treatments of advanced HCC [129,146]. These combination therapies of anti-PD-1 and anti-angiogenesis antibodies in particular may exhibit a more promising anti-HCC efficacy by both downregulating tumor angiogenesis and limiting the activity of immunosuppressive cells in the tumor microenvironment, hence promoting cytotoxic T cells to re-express their antitumor effect [181]. Importantly, a great number of new drugs, as well as new combinations of drugs with known efficacy, are currently under investigation and need to be tested in phase III clinical trials in order to verify their therapeutic safety and efficacy in HCC patients. This great interest from the scientific community in these newly developed agents is indicated by the more than 750 ongoing registered clinical trials on antineoplastic agents in HCC [https://clinicaltrials.gov (accessed on 13 November 2022)].

Another therapeutic option that could affect the OS of patients with HCC is the concept of neoadjuvant therapy. In a study, the use of dovitinib, an anti-angiogenesis factor, showed promising results [182]. More such studies are needed to further elaborate this concept of therapy that could change the future of HCC treatment. Similarly, the synchronous use of TACE and systemic treatments also shows favorable results [104,105,106,143,149].

In the field of immunotherapy, there is still a lot to discover too. Immunotherapy has offered a lot in the OS of patients with melanoma [183], and it is also part of the treatment of cancers such as pancreatic cancer and cholangiocarcinoma [167,184]. One of the most promising fields of immunotherapy, and yet quite unexplored, regards CAR-T therapy. Despite the success story of CAR-T therapy in hematologic malignancies, in solid tumors there are some limitations, such as the variety of the tumor-associated antigens and the possibly reduced penetration at the site of the tumor. However, there are promising results in HCC both in vitro and in vivo, and there are many ongoing clinical studies [184]. Glypican-3 (GPC3) is considered to be one of the most hopeful targets for CAR-T therapy, but this remains to be proven [185,186,187], while there are also promising results regarding the anti-CD133 CAR-T therapy as well [180].

In addition, the inhibition of other pathways of carcinogenesis, such as the TGF-β-associated pathway, a pathway not sufficiently studied in clinical trials, may also offer alternative treatment options [152,153,186], while the molecular enhancement of already used drugs may offer extra therapeutic options. Donafenib, a deuterated sorafenib derivative, with a more favorable safety profile, has been approved in China for the treatment of HCC [108]. There are also ongoing clinical studies investigating the safety and efficacy of donafenib in combination with anti-PD1 agents (NCT04612712, NCT04503902).

Another critical point in the treatment of advanced HCC regards the treatment after intolerance or failure of the first-line treatment. Despite the abundance of possible agents, few have been proven to be efficient, and almost all of them were studied as second-line treatments after treatment with sorafenib. However, in the new era, a rising number of patients would have been treated with different options such as bevacizumab/atezolizumab or lenvatinib [150]. Consequently, data regarding the second-line agents after treatment with these agents need to be collected.

The evolution of targeted therapies questions the efficacy of chemotherapy agents in the treatment of advanced HCC. Possibly, in the new era, the role of chemotherapy could be re-established as part of combination treatments with other targeted agents. However, despite some favorable results [90,91,188], phase III clinical trials incorporating larger patient collectives are still required.

Another field of investigation that is still mainly in the pre-clinical stage regards specific molecular targets involved in tumorigenesis such as long non-coding RNAs (lncRNAs) and micro-RNAs (miRNAs). Several miRNAs and lncRNAs have been found to be part of tumorigenesis in HCC [189,190,191] and could possibly serve as therapeutic targets, while some miRNAs could also serve as therapeutic agents. However, the related clinical studies are still very few in number and with possible safety issues [192]. The wide use of mRNA vaccines during the COVID pandemic could, however, accelerate this research.

In the era of countless possible therapeutic choices, making the choice of the best treatment for each patient is demanding. Although much progress has been made in this direction, the only established knowledge regards the favorable results of ramucirumab in patients with AFP ≥400 ng/mL [128,129]. Other possible biomarkers that could predict the response to treatment include the c-met expression for treatment with tivantinib [120,121] and TGF-β1 for treatment with inhibitors of the TGF-β receptor [153]. A recent study has recognized five biomarkers (ANG-1, cystatin B, LAP TGFb1, LOX-1, MIP-1a) related to the OS and TTP after treatment with regorafenib [193]. Further studies are needed, however, to establish such connections.

Another factor that possibly plays a role in the therapeutic response is the etiology of HCC. It seems that the benefit to the OS with sorafenib treatment is greater in HCV-positive patients, while this benefit may not exist or may be smaller in HBV-positive patients. This could also be a reason why clinical trials with sorafenib in the East, in which many HBV-positive patients were included, showed mediocre results [89,194,195]. Despite this, donafenib, a modified derivative of sorafenib, has been found to improve OS in a phase III clinical trial including 594 patients with HBV-related HCC (594/659 (90%)) [108], thus representing an efficient alternative to sorafenib in such patients.

On the other hand, immunotherapy may not be beneficial for NASH-related HCC. A recent meta-analysis of clinical trials with immunotherapy agents has shown no benefit in OS for non-viral HCC [196]. A possible theory that may explain this reduced efficacy of immunotherapy in NASH-related HCC claims that there are a lot of activated CD8+/PD1+ T-cells causing tissue damage and lacking immune surveillance functions [197]. This theory, however, is mainly based on data collected through pre-clinical mice models. A possible solution could be the combinational use of different immunotherapy agents, but this needs to be proved [196,197,198]. Undoubtedly, as patients with NASH-related HCC increase, the question of the efficacy of immunotherapy in such patients will need urgent answers.

Last but not least, the majority of clinical trials include patients with preserved hepatic function, mainly with Child–Pugh A or B7. As a result, the data regarding the rest of the patients remain limited. In a phase IV non-interventional study (GIDEON), the safety of sorafenib was assessed. Patients with Child–Pugh A had a longer period of treatment, as well as less serious or treatment-related AEs, in comparison with patients with Child–Pugh B. The duration of treatment was also less in patients with a worse BCLC stage. However, the overall safety profile was comparable among different Child–Pugh and BCLC subgroups [199,200]. In a meta-analysis regarding patients with Child–Pugh B treated with sorafenib, no significant difference was found in the prevalence of serious AEs, treatment-related deaths, or treatment discontinuation. Child–Pugh B was related to a worse OS [201]. In another subanalysis of the SHARP trial, elevated baseline concentrations of alanine aminotransferase/aspartate aminotransferase or bilirubin did not affect the safety profile of sorafenib, though they were associated with a shorter OS [202]. Consequently, sorafenib could be used, though with caution, in patients with impaired hepatic function, while more such studies regarding the rest of the treatment options are also needed.

The improvement in the OS of patients with HCC necessitates a deeper knowledge of the pathogenesis of HCC. The new era will be mainly based on combination treatments due to the molecular complexity of HCC. Moreover, further research on the genomic alterations should be carried out, since genomic alterations are responsible for HCC initiation and progression [203]. In addition, in vitro experiments including known hepatocarcinogenic aberrations in humans need to be carried out in order to investigate new molecules targeting specific mutations. Along the same line, HCC stem cell lines should be developed for novel drugs to be tested.

## 5. Conclusions

In conclusion, HCC is the most common primary liver malignancies and one of the most common cancers. HCC has been proven to be resistant to chemotherapy, and orthotopic liver transplantation constitutes the best treatment option for the patients in a terminal stage. As a result, the development of targeted therapies has captured researchers’ attention as a therapeutic tool against HCC.

There is a variety of growth factors, which are expressed in cancer cells and in healthy cells surrounding the tumor. Since the expression of these factors is related to the expansion of the disease through various molecular pathways, various drugs that target each pathway separately or a combination of them have been investigated for HCC.

Among all the drugs evaluated, sorafenib was the first to lead to promising results and was the standard of care until the combination of atezolizumab with bevacizumab proved to be superior. At the same time, many other drugs have also proved their efficacy and gained approval for the treatment of HCC, or their promising efficacy is still under evaluation. This review provides an extended and accurate update on novel potential therapeutic approaches to HCC and places particular attention on newly established biological drugs. Although there are several recent studies that have already been published on current HCC treatments [204,205,206,207,208], the present work not only summarizes the mechanisms of action of the reviewed agents but also discusses the advantages and drawbacks of each therapeutic approach and, most importantly, compares the features of newly developed agents with those of well-established treatment regimes.

It is worth mentioning that the current knowledge on the management of HCC should be the key for further studies to be carried out in order to provide insights into future perspectives and lead to the development of new and advanced drugs.

## Figures and Tables

**Figure 1 ijms-23-14117-f001:**
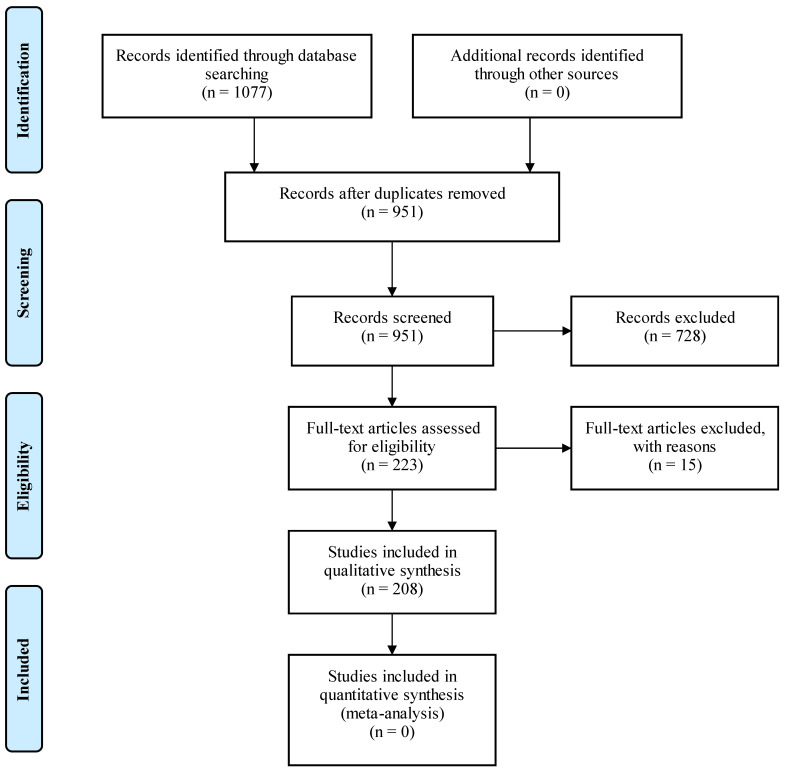
PRISMA flow diagram for the current study.

**Figure 5 ijms-23-14117-f005:**
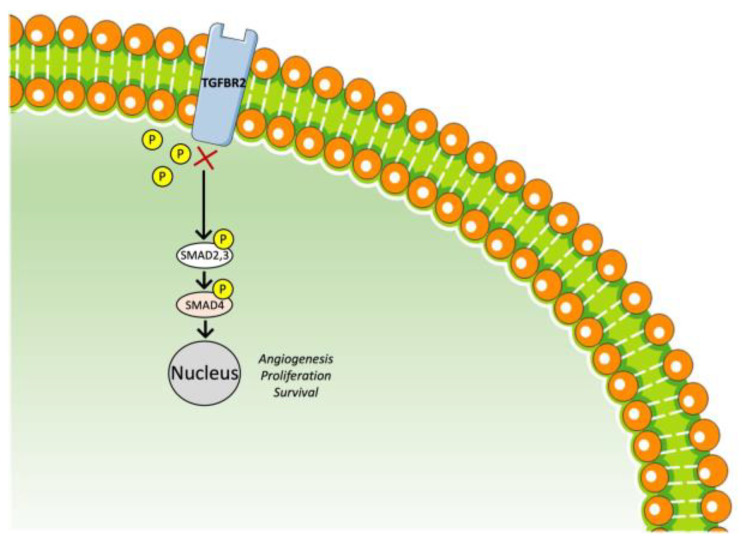
Mechanism of action of galunisertib for the treatment of HCC.

**Figure 7 ijms-23-14117-f007:**
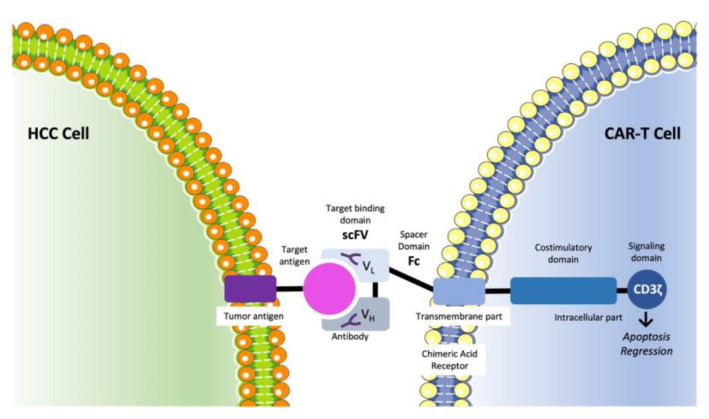
Mechanism of action of CAR-T cell therapy for the treatment of HCC.

**Table 1 ijms-23-14117-t001:** Phase III trials that evaluated targeted agents as first-line treatment.

Study				Agent	N	OS or PE	HR	*p* Value	Result
Name		Author	Year						
1	SHARP	Llovet et al. [82]	2008	*Sorafenib vs. placebo*	602	10.7 vs. 7.9	0.69 (0.55–0.87)	<0.001	Confirmed efficacy
2	ASIA-PACIFIC	Cheng et al. [83]	2009	*Sorafenib vs. placebo*	271	6.5 vs. 4.2	0.68 (0.50–0.93)	0.014	Confirmed efficacy
3	SUN	Cheng et al. [101]	2013	*Sunitinib vs. Sorafenib*	1074	7.9 vs. 10.2	1.30 (1.13–1.50)	0.0014	Negative
4	SEARCH	Zhu et al. [51]	2015	*Sorafenib plus Erlotinb vs. Sorafenib plus placebo*	720	9.5 vs. 8.5	0.93 (0.78–1.11)	0.408	Negative
5	LIGHT	Cainap et al. [67]	2015	*Linifanib vs. Sorafenib*	1035	9.1 vs. 9.8	1.04 (0.896–1.221)	-	Negative
6	REFLECT	Kudo et al. [95]	2018	*Lenvatinib vs. Sorafenib*	954	13.6 vs. 12.3	0.92 (0.79–1.06)	-	Confirmed efficacy
7	IMBRAVE150	Finn et al. [145]	2020	*Bevacizumab plus Atezolizumab vs. Sorafenib*	501	12 m OS: 67.2% vs. 54.6%	0.58 (0.42–0.79)	<0.001	Confirmed efficacy
8	-	Qin et al. [108]	2021	*Donafenib vs. Sorafenib*	659	12.1 vs. 10.3	0.831 (0.699–0.988)	0.0245	Confirmed non-inferiority as well as superiority of Donafenib
9	ORIENT-32	Ren et al. [178]	2021	*Sintilimab plus Bevacizumab*	571	Not reached vs. 10.4	0.57 (0.43–0.75)	<0.0001	Confirmed efficacy
10	CheckMate 459	Yau et al. [172]	2022	*Nivolumab vs. Sorafenib*	743	16.4 vs. 14.7	0.85 (0.72–1.02)	0.075	No superiority of Nivolumab

N: number of patients; OS: overall survival; PE: primary endpoint.

**Table 2 ijms-23-14117-t002:** Phase III trials that evaluated targeted agents as second-line treatment.

Study				Agent	N	OS or PE	HR	*p* Value	Result
Name		Author	Year						
1	METIV-HCC	Santoro et al. [117]	2013	*Tivantinib vs. placebo*	340	8.4 vs. 9.1	0.97 (0.75–1.25)	0.81	Negative
2	REACH	Zhu et al. [128]	2015	*Ramucirumab vs. placebo*	565	9.2 vs. 7.6	0.87	0.14	No superiority of Ramucirumab
3	RESORCE	Bruix et al. [101]	2017	*Regorafenib vs. placebo*	573	10.6 vs. 7.8	0.63 (0.50–0.79)	<0.001	Confirmed efficacy
4	CALESTIAL	Abou-Alfa et al. [112]	2018	*Cabozantinib vs. placebo*	707	20.2 vs. 8.0	0.76 (0.63–0.92)	0.0049	Confirmed efficacy
5	REACH-2	Zhu et al. [129]	2019	*Ramucirumab vs. placebo*	295	8.5 vs. 7.3	0.71 (0.531–0.949)	0.0199	Confirmed efficacy
6	JET-HCC	Kudo et al. [118]	2020	*Tivantinib vs. placebo*	195	10.3 vs. 8.5	0.82 (0.58–1.15)	0.082	Negative
7	KEYNOTE-240	Finn et al. [145]	2020	*Pembrolizumab vs. placebo*	413	13.9 vs. 10.6	0.781 (0.6–0.998)	0.238	Efficacy not confirmed in terms of OS
8	AHELP	Qin et al. [146]	2021	*Apatinib vs. placebo*	393	8.7 vs. 6.8	0.785 (0.617–0.998)	0.048	Confirmed efficacy

N: number of patients; OS: overall survival; PE: primary endpoint.

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
