# Peer review of "Targeted Therapies for Hepatocellular Carcinoma Treatment: A New Era Ahead—A Systematic Review"

_ijms, 2022, doi:10.3390/ijms232214117_

Round 1
Reviewer 1 Report
In the present study, Dimitroulis et al. reviewed the current knowledge of targeted therapies of HCC. The manuscript is comprehensive and informative for readers who are interested in this field. Minor revision is desirable to help readers understand the overview.
1) The image of Figure 1 is missing. There is only its title.
2) It is difficult to comprehensively understand the relationship between drugs shown here and their action targets (pathways and/or molecules). A table and a figure are desirable to summarize this information so that readers can understand it easily.
Author Response
Dear reviewers
thank you very much for your comments. Please find attached the responses. Don't hesitate to contact with me if any other information is required. Best regards.
Damaskos C, MD, MSc, PhD

Reviewer 2 Report
Dear all,
Thank you for the opportunity to review the manuscript entitled "Targeted therapies for hepatocellular carcinoma treatment: A new era in front" by Dimitroulis et al.
The authors provide a review of demographics, etiology and current targeted therapy of hepatocellular carcinoma (HCC). They conclude that the combination of atezolizumab with bevacizumab is superior to other current treatments, and also that other drugs are under current evaluation.
The authors are to be commended for their extensive effort to provide a comprehensive review on this very interesting and constantly changing topic. However, I have some major remarks.
There are several recent studies that have been already published that analyze current HCC treatments. I recommend that the authors elaborate the novelty of their manuscript and the new information their study will add to the currently knowledge. For example, the following publications should be taken into consideration:
§ Huang et al. Targeted therapy for hepatocellular carcinoma. Signal Transduct Target Ther. 2020 Aug 11;5(1):146.
§ Niu et al. Advances of Targeted Therapy for Hepatocellular Carcinoma. Front Oncol. 2021 Jul 26;11:719896.
§ Zhang et al. Recent advances in systemic therapy for hepatocellular carcinoma. Biomark Res. 2022 Jan 9;10(1):3.
§ Tella et al. Systemic therapy for advanced hepatocellular carcinoma: targeted therapies. Chin Clin Oncol. 2021 Feb;10(1):10.
§ Stotz et al. Molecular Targeted Therapies in Hepatocellular Carcinoma: Past, Present and Future. Anticancer Res. 2015 Nov;35(11):5737-44.
I also suggest to provide figures explaining the chronological discovery of the targeted and immunotherapeutic drugs and their possible pharmacologic mechanisms. This would strengthen the conclusion of the current review.
I encourage the authors to discuss ongoing clinical trials on newly discovered targeted and immunotherapeutic drugs and to discuss combination therapies more extensively.
The authors have reported that 206 articles were included in quantitative meta-analysis (Figure 1). They should explain to which quantitative analysis they refer to, report the results of this analysis and provide descriptive data (overall number of patients, demographic characteristics of the included patients, etc.).
Author Response
Thank you for the opportunity to review the manuscript entitled "Targeted therapies for hepatocellular carcinoma treatment: A new era in front" by Dimitroulis et al. The authors provide a review of demographics, etiology and current targeted therapy of hepatocellular carcinoma (HCC). They conclude that the combination of atezolizumab with bevacizumab is superior to other current treatments, and also that other drugs are under current evaluation. The authors are to be commended for their extensive effort to provide a comprehensive review on this very interesting and constantly changing topic. However, I have some major remarks.
Comments:
- There are several recent studies that have been already published that analyze current HCC treatments. I recommend that the authors elaborate the novelty of their manuscript and the new information their study will add to the currently knowledge. For example, the following publications should be taken into consideration:
- Huang et al. Targeted therapy for hepatocellular carcinoma. Signal Transduct Target Ther. 2020 Aug 11;5(1):146.
- Niu et al. Advances of Targeted Therapy for Hepatocellular Carcinoma. Front Oncol. 2021 Jul 26;11:719896.
- Zhang et al. Recent advances in systemic therapy for hepatocellular carcinoma. Biomark Res. 2022 Jan 9;10(1):3.
- Tella et al. Systemic therapy for advanced hepatocellular carcinoma: targeted therapies. Chin Clin Oncol. 2021 Feb;10(1):10.
- Stotz et al. Molecular Targeted Therapies in Hepatocellular Carcinoma: Past, Present and Future. Anticancer Res. 2015 Nov;35(11):5737-44.
Response:
We would like to thank the reviewer for this constructive comment. In the conclusion, we have now elaborated on the novelty of our manuscript and the new information our study will add to the current knowledge, by taking into consideration the proposed publications.
- I also suggest to provide figures explaining the chronological discovery of the targeted and immunotherapeutic drugs and their possible pharmacologic mechanisms. This would strengthen the conclusion of the current review.
Response:
Thank you for this useful suggestion. We have now added figures summarizing the relationship between the reviewed drugs and their action targets.
- I encourage the authors to discuss ongoing clinical trials on newly discovered targeted and immunotherapeutic drugs and to discuss combination therapies more extensively.
Response:
Thank you for this suggestion. In the discussion, we now discuss ongoing clinical trials on newly discovered targeted and immunotherapeutic drugs, and combination therapies more extensively.
- The authors have reported that 206 articles were included in quantitative meta-analysis (Figure 1). They should explain to which quantitative analysis they refer to, report the results of this analysis and provide descriptive data (overall number of patients, demographic characteristics of the included patients, etc.).
Response:
We would like to thank the reviewer for this comment. As a matter of fact, the missing Figure 1 on the selection process undoubtedly caused confusion to the readers. By adding the missing diagram and more extensively explaining the results of our analysis in the Materials and Methods, we now hope to have addressed the reviewer’s comment.
Reviewer 3 Report
This article is an extended and accurate uptodate on new potential therapeutic approach to liver cancer
that is a most common cancer worldwide, accounting for about 6% of new cancer cases, representing the third most common cause of cancer-related
death. The authors place a particular attention to new biologocal drugs, comparing then with sorafenib.
The article is well written and well organized and gives a real uptodate to therapeutic approach of hepatocarcinoma.
Just an aspect that I do not share. Some paragraph are too long and some (i.e., Liver lobule: The organization of
hepatic parenchyma ) are not necessary.
Author Response
This article is an extended and accurate up to date on new potential therapeutic approach to liver cancer that is a most common cancer worldwide, accounting for about 6% of new cancer cases, representing the third most common cause of cancer-related death. The authors place a particular attention to new biological drugs, comparing them with sorafenib.
The article is well written and well organized and gives a real up to date to therapeutic approach of hepatocarcinoma.
Just an aspect that I do not share. Some paragraphs are too long and some (i.e., Liver lobule: The organization of hepatic parenchyma) are not necessary.
Response:
Thank you for this useful comment. Despite the fact that some paragraphs are long and others are not necessary, we strongly believe that they are useful in order to understand the basis and the mechanism of the described treatments.
Round 2
Reviewer 2 Report
Dear all,
the authors have resubmitted a revised and improved version of their manuscript. However, the authors should provide references for all Figures except Figure 1 and check the manuscript for minor language errors and typos.
Best regards!
Author Response
Dear Reviewer,
thank you very much again for your comments. I would like to let you know that we designed the figures of our manuscript, according to its mechanism as it is described. So, the reference of its figure is the same with the text's, as indicated. In order to facilitate your work, please find attached the following table. Regarding the language and typographical errors, we checked again the manuscript. However, we accept every suggested correction. Don't hesitate to contact with me if any other information is required. Thank you very much again.
Damaskos C, MD, MSc, PhD
NS Christeas, Laboratory of Experimental Surgery and Surgical Research, Medical School, National and Kapodistrian University of Athens, Greece
